# Tractable Optimality in Episodic Latent MABs

**Jeongyeol Kwon**
University of Wisconsin-Madison
jeongyeol.kwon@wisc.edu

**Yonathan Efroni**
Meta, New York
jonathan.efroni@gmail.com

**Constantine Caramanis**
The University of Texas at Austin
constantine@utexas.edu

**Shie Mannor**
Technion, NVIDIA
shie@ee.technion.ac.il,
smannor@nvidia.com

## Abstract

We consider a multi-armed bandit problem with $A$ actions and $M$ latent contexts, where an agent interacts with the environment for an episode of $H$ time steps. Depending on the length of the episode, the learner may not be able to estimate accurately the latent context. The resulting partial observation of the environment makes the learning task significantly more challenging. Without any additional structural assumptions, existing techniques to tackle partially observed settings imply the decision maker can learn a near-optimal policy with $O(A)^H$ episodes, but do not promise more. In this work, we show that learning with *polynomial* samples in $A$ is possible. We achieve this by using techniques from experiment design. Then, through a method-of-moments approach, we design a procedure that provably learns a near-optimal policy with $O(\text{poly}(A) + \text{poly}(M, H)^{\min(M,H)})$ interactions. In practice, we show that we can formulate the moment-matching via maximum likelihood estimation. In our experiments, this significantly outperforms the worst-case guarantees, as well as existing practical methods.

## 1 Introduction

In Multi-Armed Bandits (MABs), an agent learns to act optimally by interacting with an unknown environment. In many applications, interaction sessions are short, relative to the complexity of the overall task. As a motivating example, we consider a large content website that interacts with users arriving to the site, by making sequential recommendations. In each such *episode*, the system has only a few chances to recommend items before the user leaves the website – an interaction time typically much smaller than the number of actions or the types of users. In such settings, agents often have access to short horizon episodes, and have to learn how to process observations from different episodes to learn the best adaptation strategy.

Motivated by such examples, we consider the problem of learning a near-optimal policy in Latent Multi-Armed Bandits (LMABs). In each episode with time-horizon $H$, the agent interacts with one of $M$ possible MAB environments (e.g., type of user) randomly chosen by nature. Without knowing the identity of the environment (we call this the *latent context*), an agent aims to maximize the expected cumulative reward per episode (see Definition 2.1 for a formal description). The LMAB framework is different from the setting considered in [31] where multiple episodes proceed in parallel without limiting the horizon of an episode $H$. For long horizons $H$, we show that it is possible to find a near-optimal policy and to determine near-optimal actions for each episode, as if we knew the hidden context. If $H \ll A$ where $A$ is the total number of actions, however, this is no longer possible. Instead, we aim to learn the best *history-dependent* policy for $H$ time steps.

36th Conference on Neural Information Processing Systems (NeurIPS 2022).

## 1.1 Our Results and Contributions

In this work, we study the problem of learning a near optimal policy of an LMABs for both long and short horizon $H$. In the long horizon case, we show the problem's latent parameters are learnable. However, for short horizon, the latent parameters may not be identifiable and a different perspective is required.

A naive approach to learn a near optimal policy of an LMAB starts by casting the problem as a Markov Decision Process (MDP). Assume that the reward values are discrete and defined over a finite alphabet of cardinality $Z$. By defining the state space as the set of all sequences of history observations an LMAB can be formulated as an MDP with $O((AZ)^H)$ states. Appealing to standard reinforcement learning algorithms (*e.g.,* [18, 6]) we can learn an $\epsilon$-optimal policy with $O(AZ)^H/\epsilon^2$ samples.

A natural way to improve upon the naive approach is through using the unique structure of the LMAB setting. In this work we focus on the following question: *can we learn a near-optimal policy with fewer than $O(AZ)^H/\epsilon^2$ samples as the naive approach?*

Our work answers the above question affirmatively. Specifically, our main contributions are as follows. We show that the dependence of our algorithm is $\text{poly}(A) + \text{poly}(H, M)^{\min(H,M)}$, and thus tractable when either $H$ or $M$ is small, even for very large $A$. That is, we are particularly focused on the setting with a few contexts or relatively short episodes (in comparison to $A$), *i.e.,* $M = O(1)$ or $H = O(1)$, where a natural objective is to learn a near-optimal history-dependent policy for $H$ time steps (see also Figure 3 in Appendix B.1).

## 1.2 Related Work

Due to space constraints, we discuss only the most closely related work here, and defer a lengthier discussion on related work to Appendix B.1.

**Learning priors in multi-armed bandit problems**  Several recent works have considered the Bayesian learning framework with short time-horizon [29, 35, 21]. The focus in this line of work is on the design of algorithms that learn the prior, while acting with a fixed Bayesian algorithm (e.g., Thompson sampling). While Bayesian learning with short time-horizon may be viewed as a special case of LMABs, the baseline policy we compare ourselves to is the optimal $H$-step policy, which is a harder baseline than considering a fixed Bayesian algorithm.

**Latent MDPs**  Some prior work considers the framework of Latent MDPs (LMDPs), which is the MDP generalization of LMAB [16, 37, 24, 23]. In particular, [24] has shown the information-theoretic limit of learning in LMDPs with no assumptions on MDPs, *i.e.,* an exponential number of sample episodes $\Omega(A^M)$ is necessary to learn a near-optimal policy in LMDPs. In contrast, we show that in LMABs, the required number of episodes can be polynomial in $A$. This does not contradict the result in [24], since their lower bound construction comes from the challenge in state-exploration with latent contexts. In contrast, there is no state-exploration issue for bandits, which enables the polynomial sample complexity in $A$. Furthermore, our upper bound does not require any assumptions or additional information such as good initialization or separations as in [24]. To the best of our knowledge, no existing results are known for learning a near optimal policy of LMAB instances for $M \geq 3$ without further assumptions.

**Learning in POMDPs with full-rank observations**  One popular learning approach in partially observable systems is the tensor-decomposition method, which extracts the realization of model parameters from third-order tensors [1, 7, 15]. However, the recovery of model parameters require specific geometric assumptions on the full-rankness of a certain test-observation matrix. Furthermore, most prior work requires a uniform reachability assumption, *i.e.,* all latent spaces should be reached with non-negligible probabilities by any exploration policy for the parameter recovery. Recent results in [19, 30] have shown that the uniform reachability assumption can be dropped with the optimism principle. However, they still require the full-rankness of a test-observation matrix to keep the volume of a confidence set explicitly bounded. Since LMAB instances do not necessarily satisfy the full-rankness assumption, their results do not imply an upper bound for learning LMABs.

## 2 Preliminaries

We define the problem of **episodic** latent multi-armed bandits with time-horizon $H \geq 2$ as follows:

**Definition 2.1 (Latent Multi-Armed Bandit (LMAB))** *Let LMAB be a tuple $\mathcal{B} = (\mathcal{A}, \{w_m\}_{m=1}^M, \{\mu_m\}_{m=1}^M)$, where $\mathcal{A}$ is a set of actions, $\{w_m\}_{m=1}^M$ are the mixing weights such that a latent context $m$ is randomly chosen with probability $w_m$, and $\mu_m$ is the model parameter that describes a reward distribution, i.e., $\mathbb{P}_{\mu_m}(r \mid a) := \mathbb{P}(r \mid m, a)$, according to an action $a \in \mathcal{A}$ conditioning on a latent context $m$ (each $\mu_m$ parameterizes the probability model, and is not necessarily a mean reward vector).*

We do not assume a priori knowledge of mixing weights. The bulk of this paper considers discrete reward realizations, when the support of the reward distribution is finite and bounded. In Appendix D, we also show that our results can be adapted to Gaussian reward distributions.

**Assumption 2.2 (Discrete Rewards)** *The reward distribution has finite and bounded support. The reward attains a value in the set $\mathcal{Z}$. We assume that for all $z \in \mathcal{Z}$ we have $|z| \leq 1$. We denote the cardinality of $\mathcal{Z}$ as $Z$ and assume that $Z = O(1)$.*

As an example, Bernoulli distribution satisfies Assumption 2.2 with $\mathcal{Z} = \{0, 1\}$ and $Z = 2$. We denote the probability of observing a reward value $z$ by playing an action $a$ as $\mu_m(a, z) := \mathbb{P}(r = z \mid m, a)$ in a context $m$. We often use $\mu_m$ as a reward-probability vector in $\mathbb{R}^{AZ}$ indexed by a tuple $(a, z) \in \mathcal{A} \times \mathcal{Z}$.

At the beginning of every episode, a latent context $m \in [M]$ is sampled from a mixing distribution $\{w_m\}_{m=1}^M$ and fixed for $H$ time steps, however we cannot observe $m$ directly. We consider a policy class $\Pi$ which contains all history-dependent policies $\pi : (\mathcal{A} \times \mathcal{Z})^* \to \mathcal{A}$. Our goal is to find a near optimal policy $\pi \in \Pi$ that maximizes the expected cumulative reward $V$ for each episode $V^\star = \max_{\pi \in \Pi} V(\pi) := \mathbb{E}^\pi \left[ \sum_{t=1}^H r_t \right]$, where the expectation is taken over latent contexts and rewards generated by an LMAB instance, and actions following a policy $\pi$.

**Definition 2.3 (Approximate Planning Oracle)** *A planning oracle receives an LMAB instance $\mathcal{B}$ and returns an $\epsilon$-approximate policy $\pi$ such that $V^\star - V(\pi) \leq \epsilon$.*

Concretely, the point-based value-iteration (PBVI) algorithm [33] is an $\epsilon$-approximate planning algorithm which runs in time $O(HMAZ(H^2/\epsilon)^{O(M)})$.

**Additional notation.** For any quantity $q$ with respect to the true LMAB, $\mathcal{B}$, we use $\hat{q}$ to refer to its corresponding empirical estimate. We use $w_{\min} := \min_{m \in [M]} w_m$ for the minimum mixing weight. For any $l^{th}$-order tensor $T_l \in \mathbb{R}^{n \times \cdots \times n}$ ($n$ repeated $l$ times), we denote $\|T_l\|_\infty$ for the element-wise largest absolute value. For any vector in $v \in \mathbb{R}^n$ in dimension $n \in \mathbb{N}_+$, we use $v^{\otimes l}$ to denote a degree $l$ tensorization of $v$. We often use $\|\mu_m(a, \cdot) - \hat{\mu}_m(a, \cdot)\|_1$ to mean the $l_1$ statistical distance between reward distributions: $\sum_{z \in \mathcal{Z}} |\mu_m(a, z) - \hat{\mu}_m(a, z)|$. Lastly, we denote by $a_t$ and $r_t$ as the action and reward realizations observed at time step $t \in [H]$.

## 3 Sample-Complexity of Learning LMABs

In this section, we develop our main algorithm that learns a near-optimal policy of an LMAB instance with $O(\text{poly}(A) + \text{poly}(M, H)^{\min(M,H)})$ samples. Towards achieving this goal, we first elaborate on a low rank representation of the LMAB problem. Then, we show how experimental design techniques can assist in utilizing this structure to improve over the naive upper bound to the problem.

### 3.1 Dimensionality Reduction via Experimental Design

When $H = 1$, estimating the latent context is not possible, and executing a standard MAB strategy (e.g., UCB) is optimal. However, to obtain a near-optimal history-dependent policy with longer time-horizons $H > 1$, tracking the mean reward of an action is not enough, since rewards from previous time steps are correlated with current and future rewards.

We consider a model-based learning approach from the $l^{\text{th}}$-order moments of reward observations. Since there are $A$ actions, this implies that we have $O(A^l)$ quantities to estimate, which would incur $O(A^l)$ sample complexity if we separately measure every correlation between $l$ pairs of actions. On the other hand, when $M \ll A$, the distributions $\mu_m$ (recall these are reward probabilities for each action, in each context) occupy only a $M$-dimensional subspace of $AZ$-dimensional space: $\mathbf{U} := \text{span}\{\mu_1, \mu_2, \ldots, \mu_M\}$. Thus if we could estimate $\mathbf{U}$, and project all observations to $\mathbf{U}$, we could remove the unfavorable dependence on $A$.

However, even when the subspace $\mathbf{U}$ is known a priori, dimensionality reduction with bandit feedback is non-trivial. We need to compute directly the projection onto $\mathbf{U}$ of estimates of the reward probability vectors. To see the challenge, let $\{\beta_j\}_{j=1}^M \subset \mathbb{R}^{AZ}$ be an orthonormal basis of $\mathbf{U}$. To compute the projected estimate $\mu_m^\top \beta_j$, we need a sampling policy $\pi$ such that $\pi(a) \propto \sum_z |\beta_j(a, z)|$. The variance of this sampling policy can be as much as $O(\|\beta_j\|_1^2)$, which in general scales with $\|\beta_j\|_1^2 = O(A)$. Therefore, reliable estimation for any statistics in the reduced dimension would still have a dependence on $A$. In particular, since our approach is based on higher-order method-of-moments, an estimation of $l^{\text{th}}$-order statistics for $l \geq 2$ would require $\Omega(A^{l/2})$ samples.

The general idea of dimensionality reduction is not fatally flawed. Instead, we need to avoid the pitfalls of the approach outlined above. First, the calculation above shows that we need to control the $\|\cdot\|_1$-norm of the vectors $\beta_j$, ideally, $\|\beta_j\|_1 = O(1)$. Second, we only need a good estimate for each $\mu_m$ such that $\max_{a \in \mathcal{A}} \|\mu_m(a, \cdot) - \hat{\mu}_m(a, \cdot)\|_1 \leq \epsilon$ to compute an $\epsilon$-optimal policy.

The key is to show that we can choose $\beta_j$'s to be a subset of the standard basis in $\mathbb{R}^{AZ}$ (hence they will have $\|\beta_j\|_1 = 1$), in such a way that guarantees the approximation quality for $\mu_m$ from estimating $\mu_m^\top \beta_j$'s. In terms of the original problem, the existence of such a subset of the standard basis is equivalent to the existence of a small set of informative action-value pairs, that are sufficiently correlated with all other action-value pairs. This is called a *core set* in the experimental design literature, and its existence in our context, is an important consequence of the Kiefer–Wolfowitz theorem. Specifically, we can select a core set of coordinates with the following crucial lemma:

**Lemma 3.1** *Let $\mathbf{U}$ be a given $k$-dimensional linear subspace in $\mathbb{R}^d$ where $d \gg k$ and let $u, \bar{u} \in \mathbf{U}$. There exists an algorithm that runs in time $\tilde{O}(dk^2)$ and returns the following.*

1. *A core set of at most $n = 4k \log \log k + 16$ coordinates $\{i_j\}_{j=1}^n \subseteq [d]$ such that $\|u - \bar{u}\|_\infty \leq \sqrt{2k} \max_{j \in [n]} |u(i_j) - \bar{u}(i_j)|$.*

2. *A linear transformation that maps $[\bar{u}(i_1), \ldots, \bar{u}(i_n)]$ to its corresponding $\bar{u} \in \mathbf{U}$.*

Note that in our setting, $d = AZ$ and $k = M$. We prove Lemma 3.1 in Section C.1, essentially as a corollary of the Kiefer–Wolfowitz theorem (and its geometric interpretation) for (near)-optimal experimental design [20, 39].

In the context of bandits and RL, the Kiefer–Wolfowitz theorem has been used to study how the misspecification in linear representation changes the problem landscape (*e.g.,* [27, 32]). However, experimental design has not been previously used for dimensionality reduction for problems with bandit feedback (as far as we know). We believe it is a powerful tool. We review the basics of experimental design in Appendix B.2.

A direct consequence of Lemma 3.1 is an algorithm to find a set of coordinates of size $\tilde{O}(M)$ that are sufficient to reconstruct the latent reward model $\{\mu_m\}_{m=1}^M$ where $\mu_m \in \mathbb{R}^{AZ}$. We refer to this set of coordinates as the set of core action-value pairs.

**Corollary 3.2** *Suppose the subspace $\mathbf{U} = \text{span}\{\mu_1, \ldots, \mu_M\}$ is given. Then for any $\hat{\mu} \in \mathbf{U}$, there exists an algorithm that runs in time $\tilde{O}(ZAM^2)$ and returns the following.*

1. *A set of core action-value pairs $\{(a_j, z_j)\}_{j=1}^n \subseteq \mathcal{A} \times \mathcal{Z}$ of size at most $n = 4M \log \log M + 16$, such that for all $m \in [M]$*

$$\max_{a \in \mathcal{A}} \|\mu_m(a, \cdot) - \hat{\mu}_m(a, \cdot)\|_1 \leq 2Z\sqrt{2M} \cdot \max_{j \in [n]} |\nu_m(j) - \hat{\nu}_m(j)|,$$

*where $\nu_m, \hat{\nu}_m \in \mathbb{R}^n$ such that $\nu_m(j) := \mu_m(a_j, z_j)$ and $\hat{\nu}_m(j) := \hat{\mu}_m(a_j, z_j)$.*

2. *A linear transformation that maps $[\hat{\nu}(i_1), \ldots, \hat{\nu}(i_n)]$ to its corresponding $\hat{\mu} \in \mathbf{U}$.*

Assuming access to the subspace $\mathbf{U}$ (we remove this assumption in Section 3.4), Corollary 3.2 implies a strategy for estimating the latent model parameters $\{\mu_m\}_{m=1}^M$: obtain core action-value pairs $\{(a_j, z_j)\}_{j=1}^n$, estimate $\{\nu_m\}_{m=1}^M$, and construct latent model parameters via a linear transformation of $\{\nu_m\}_{m=1}^M$. In the following sections, we show that by matching higher-order moments we can estimate $\{\nu_m\}_{m=1}^M$ to sufficiently good accuracy with only polynomial dependence in $A$.

## 3.2 $H \geq 2M - 1$: Identifiable Regime in Wasserstein Metric

In the regime where that $H \geq 2M - 1$ we can measure moments up to order $(2M - 1)^{\text{th}}$. Then, we can leverage recent advances in learning parameters of mixture distributions from higher-order moments [40, 12]. That is, we can recover $\{\nu_m\}_{m=1}^M$ by estimating $2M - 1$ higher-order moments and find $\{\hat{\nu}_m\}_{m=1}^M$ that matches them. This further emphasizes the importance of the dimensionality reduction we take to obtain the set of core action-values pairs; otherwise, a moment-based approach to recover $\{\mu_m\}_{m=1}^M$ would have an exponential dependence in $A$.

We now elaborate on the estimation procedure of $\{\nu_m\}_{m=1}^M$ from higher-order moments. We define the $l$-order tensor as $T_l := \sum_{m=1}^M w_m \nu_m^{\otimes l}$. For an LMAB instance, we can access the tensor $T_l$ using observational data by simply estimating correlations between $l$ core action-value pairs within each episode. Specifically, for any $\mathcal{I} = (i_1, i_2, ..., i_l) \in [n]^l$ the tensor $T_l(i_1, i_2, ..., i_l)$ is also given by

$$T_l(i_1, i_2, ..., i_l) = \mathbb{E}^{\pi_\mathcal{I}} \left[ \Pi_{t=1}^l \mathbb{1}\{r_t = z_{i_t}\} \right], \tag{1}$$

where $\pi_\mathcal{I}$ is a policy that performs the sequence of actions $(a_{i_1}, a_{i_2}, ..., a_{i_l})$ for $t = 1, \ldots, l$.

Suppose we find estimators $\{(\hat{w}_m, \hat{\nu}_m)\}_{m=1}^M$ such that moments match $T_l$ up to error $\delta$ for all $l \in [2M - 1]$. The results in [40, 12] imply that the closeness in moments of up to $(2M - 1)^{\text{th}}$ degree implies the closeness in model parameters:

**Lemma 3.3** *Suppose $\|T_l - \sum_{m=1}^M \hat{w}_m \hat{\nu}_m^{\otimes l}\|_\infty < \delta$ for all $l = 1, 2, ..., 2M - 1$ for some sufficiently small $\delta > 0$. Then,*

$$\inf_\Gamma \sum_{(m,m') \in [M]^2} \Gamma(m, m') \cdot \|\nu_m - \hat{\nu}_{m'}\|_\infty \leq O\left( M^3 n \cdot \delta^{-1/(2M-1)} \right), \tag{2}$$

*where $\Gamma$ is a joint distribution over $(m, m') \in [M]^2$ satisfying*

$$\Gamma(m, m') \in \mathbb{R}_+^{M \times M} : \sum_{m'=1}^M \Gamma(m, m') = w_m, \sum_{m=1}^M \Gamma(m, m') = \hat{w}_m. \tag{3}$$

The form of guarantee given for $\{\hat{w}_m, \hat{\nu}_m\}_{m=1}^M$ is in the *Wasserstein distance* between two latent model parameters. A useful property of the Wasserstein metric is that the distance measure is invariant to permutation of individual components, and flexible with arbitrarily small mixing probabilities or arbitrarily close components [40] (see also Appendix B.3 for the review on Wasserstein distance).

Once we obtain estimates of $\{\nu_m\}_{m=1}^M$, we can estimate $\{\hat{\mu}_m\}_{m=1}^M$ as implied by Corollary 3.2 (see Appendix C.8 for further details). We then show that for any history-dependent policy $\pi$, the expected cumulative rewards of $\pi$ are approximately the same for any close LMAB instances.

**Proposition 3.4** *Let $\mathcal{B} = (\mathcal{A}, \{w_m\}_{m=1}^M, \{\mu_m\}_{m=1}^M)$ and $\hat{\mathcal{B}} = (\mathcal{A}, \{\hat{w}_m\}_{m=1}^M, \{\hat{\mu}_m\}_{m=1}^M)$ be any two LMABs. Then, for any history-dependent policy $\pi : (\mathcal{A} \times \mathcal{Z})^* \to \mathcal{A}$, we have*

$$|V(\pi) - \hat{V}(\pi)| \leq H^2 \cdot \inf_\Gamma \sum_{(m,m') \in [M]^2} \left( \Gamma(m, m') \cdot \max_{a \in \mathcal{A}} \|\mu_m(a, \cdot) - \hat{\mu}_{m'}(a, \cdot)\|_1 \right), \tag{4}$$

*where the infimum over $\Gamma$ is taken over joint distributions over $(m, m')$ satisfying* (3).

From Corollary 3.2 we have $\max_{a \in \mathcal{A}} \|\mu_m(a, \cdot) - \hat{\mu}_{m'}(a, \cdot)\|_1 \leq 2Z\sqrt{2M}\|\nu_m - \hat{\nu}_{m'}\|_\infty$ for any $m, m' \in [M]$. Plugging this into Proposition 3.4, we have

$$|V(\pi) - \hat{V}(\pi)| \leq \text{poly}(H, Z, M, n) \cdot \delta^{-1/(2M-1)}.$$

Thus, using Lemma 3.3 with $\delta < (\text{poly}(H, Z, M, n)/\epsilon)^{2M-1}$, we can conclude that any $\epsilon$-optimal policy for $\hat{\mathcal{B}}$ is $O(\epsilon)$-optimal for the underlying LMAB $\mathcal{B}$.

### 3.3 $H < 2M - 1$: Unidentifiable Regime with Short Time-Horizon

A more interesting regime is the one in which the time-horizon $H$ is smaller than the required degree of moments $2M - 1$. For such a setting *we cannot measure moments of degree higher than $H$*. Therefore, if $H < 2M - 1$, we cannot rely on the identifiablility of the underlying LMAB model.

Instead, we make the following claim: to compute an optimal policy only for time horizon $H < 2M - 1$, we only need to match the $H^{\text{th}}$ order moments. This is formalized in the next lemma.

**Lemma 3.5** *Suppose that* $\| \sum_{m=1}^M w_m \mu_m^{\otimes H} - \sum_{m=1}^M \hat{w}_m \hat{\mu}_m^{\otimes H} \|_\infty \leq \delta$, *then for any history dependent policy* $\pi \in \Pi$, *we have*

$$|V(\pi) - \hat{V}(\pi)| \leq H Z^H \cdot \delta.$$

Therefore, it is sufficient to find estimates of mixing weights and latent models that match the measured $H^{\text{th}}$-order moment, and then compute an optimal policy for the estimated model. Lemma 3.5 is natural for discrete reward distributions with bounded support. Interestingly, we extend this result to Gaussian rewards, which are continuous and unbounded, in Appendix D.

Hence, a natural idea is to estimate parameters $\{\hat{w}_m\}_{m=1}^M, \{\hat{\mu}_m\}_{m=1}^M$ that satisfy the condition in Lemma 3.5 without incurring $O(A^H)$ sample complexity. This is possible if we have good estimates of moment-matching parameters for a set of core action-values.

**Proposition 3.6** *For any given $l \geq 1$, if* $\| \sum_{m=1}^M w_m \nu_m^{\otimes l} - \sum_{m=1}^M \hat{w}_m \hat{\nu}_m^{\otimes l} \|_\infty \leq \delta$, *then*

$$\left\| \sum_{m=1}^M w_m \mu_m^{\otimes l} - \sum_{m=1}^M \hat{w}_m \hat{\mu}_m^{\otimes l} \right\|_\infty \leq (2M)^{l/2} \cdot \delta.$$

By Proposition 3.6, we can conclude that it is sufficient to estimate $T_H := \sum_{m=1}^M w_m \nu_m^{\otimes H}$ and find $\{(\hat{w}_m, \hat{\nu}_m)\}_{m=1}^M$ that matches $T_H$ element-wise up to accuracy $\delta := (\epsilon/H)/(Z\sqrt{2M})^H$.

### 3.4 Main Result

The sections above have outlined the key ideas we need *assuming knowledge of* $\mathbf{U}$. In this section, we show how we can estimate $\mathbf{U}$, and we describe the complete procedure that learns a near-optimal policy of an LMAB instance. We give the details in Algorithm 1. Our algorithm is divided to three steps as detailed below.

**Step 1: Estimating $\mathbf{U}$.** Algorithm 1 first estimates the second-order moments $M_2 = \sum_{m=1}^M w_m \mu_m \mu_m^\top$. Let $\widehat{\mathbf{U}}$ be the top-$M$ eigenvectors of an empirical estimate of $M_2$, where $\hat{M}_2$ is constructed by collecting samples of reward correlations by taking random actions at first two time steps. Note that $\widehat{\mathbf{U}}$ may not be a good proxy for some $\mu_m$ with small mixing probability $w_m \approx 0$, and thus all elements in $\mathbf{U}$ are not necessarily close to $\widehat{\mathbf{U}}$. We defer the details of subspace recovery procedure to the proof of the following lemma in Appendix C.7.

**Lemma 3.7** *Let $\widehat{\mathbf{U}}$ be a subspace spanned by top-$M$ eigenvectors of $\hat{M}_2$. After we estimate $\hat{M}_2$ using $N_0 = O(A^4 Z^2 \log(ZA/\eta)/\delta_{\text{sub}}^4)$ episodes, with probability at least $1 - \eta$, for all $m \in [M]$, there exists $\Delta_m : \|\Delta_m\|_\infty \leq \delta_{\text{sub}}/w_m^{1/2}$ such that $\mu_m + \Delta_m \in \widehat{\mathbf{U}}$.*

Our choice of $\delta_{\text{sub}}$ differs in two regimes as the following:

$$\delta_{\text{sub}} = \frac{\epsilon}{2ZMH^2}, \qquad\qquad\qquad\qquad \text{if: } H \geq 2M - 1,$$

$$\delta_{\text{sub}} = \frac{\min\left( \sqrt{w_{\min} + \epsilon/(MH^2(Z\sqrt{2M})^H)}, \ \epsilon/(H\sqrt{M}) \right)}{2Z\sqrt{M}H}, \qquad \text{else: } H < 2M - 1, \quad (5)$$

where $w_{\min} = \min_{m \in [M]} w_m$. The main difference in two regimes is that for $H \geq 2M - 1$, when the parameters are identifiable, latent contexts with small mixing probabilities can be ignored

---

**Algorithm 1**

---

1: **Input:** Accuracy level $\epsilon, \delta_{\text{sub}}, \delta_{\text{tsr}} > 0$, model parameters $M, A, Z, H$.
2: **// Step 1**: Estimate subspace $\mathbf{U}$.
3: Construct $\hat{M}_2$, an estimate of $M_2 = \sum_{m=1}^{M} w_m \mu_m \mu_m^\top$ using $N_0 = \tilde{O}(A^4 Z^2 / \delta_{\text{sub}}^4)$ episodes.
4: Calculate $\widehat{\mathbf{U}}$, the span of top-$M$ eigenvectors of $\hat{M}_2$.
5: **// Step 2**: Estimate $\{(w_m, \nu_m)\}_{m=1}^{M}$.
6: Get core action-value pairs $\{(a_j, z_j)\}_{j=1}^{n}$ by calling Corollary 3.2.
7: Construct $\{\hat{T}_l\}_{l=1}^{\min(H, 2M-1)}$ using $N = \tilde{O}(n^{\min(2M-1,H)} \cdot M^2 / \delta_{\text{tsr}}^2)$ episodes.
8: Find valid empirical parameters $\{(\hat{w}_m, \hat{\nu}_m)\}_{m=1}^{M}$ satisfying (7).
9: **// Step 3**: Use the core-action value pairs to construct an empirical LMAB.
10: Construct empirical model $\hat{\mathcal{B}} = (\mathcal{A}, \{\hat{w}_m\}_{m=1}^{M}, \{\hat{\mu}_m\}_{m=1}^{M})$.
11: **Output:** optimal policy of $\hat{\mathcal{B}}$ by calling a planning oracle (Definition 2.3) for $\hat{\mathcal{B}}$.

---

once $w_m = o(\epsilon/M)$ to guarantee the closeness in distributions of observations. However, in the parameter unidentifiable regime, total variation distance is bounded only through errors in the moment space. Therefore, the estimated subspace needs to be accurate even for contexts with small mixing probabilities to keep the higher-order moments well approximated. Note that for instances with well-balanced mixing probabilities, *i.e.*, if $w_{\min} = \Omega(1/M)$, the order of $\delta_{\text{sub}}$ remains the same as in the $H \geq 2M - 1$ case.

**Step 2: Moment Matching**. Given $\widehat{\mathbf{U}}$, the subspace spanned by top-$M$ eigenvectors of $\hat{M}_2$, Algorithm 1, follows the procedure described in Sections 3.1-3.3. It constructs the set of (approximate) core action-values pairs $\{(a_j, z_j)\}_{j=1}^{n}$ (see Appendix C.8.1 for the detailed algorithm). Then, we construct higher-order moments of the core action-value pairs. For every multi-index $(i_1, i_2, ..., i_l) \in [n]^l$, using $N_1 = O(\log(ln^l/\eta)/\delta_{\text{tsr}}^2)$ episodes, we execute $a_t^k = a_{i_t}$ for $t = 1, ..., l$ and estimate higher-order moments (as also described in equation (1))

$$\hat{T}_l(i_1, i_2, ..., i_l) = \frac{1}{N_1} \sum_{k=1}^{N_1} \Pi_{t=1}^{l} \mathbb{1}\left\{r_t^k = z_{i_t}\right\}. \tag{6}$$

Using standard concentration inequalities and applying the union bounds over all elements in tensors, we get $\|\hat{T}_l - T_l\|_\infty < \delta_{\text{tsr}}$ with probability at least $1 - \eta$. Then we find empirical parameters $\{(\hat{w}_m, \hat{\nu}_m)\}_{m=1}^{M}$ that satisfy

$$\left\|\sum_{m=1}^{M} \hat{w}_m \hat{\nu}_m^{\otimes l} - \hat{T}_l\right\|_\infty < \delta_{\text{tsr}}, \qquad \forall l \in [\min(H, 2M - 1)]. \tag{7}$$

We set $\delta_{\text{tsr}} = O(\epsilon/(ZH^2 M^{3.5} n))^{2M-1}$ when $H \geq 2M - 1$. In the parameter unidentifiable regime $H < 2M - 1$, we set $\delta_{\text{tsr}} = O(\epsilon/H)/(Z\sqrt{2M})^H$.

**Step 3: Constructing Empirical LMAB**. Finally, Algorithm 1 uses the estimates $\{(\hat{w}_m, \hat{\nu}_m)\}_{m=1}^{M}$ to construct an empirical model $\hat{\mathcal{B}} = (\mathcal{A}, \{\hat{w}_m\}_{m=1}^{M}, \{\hat{\mu}_m\}_{m=1}^{M})$ after proper clipping and normalization. For this step to succeed, we require $\hat{w}_m$ and $\hat{\nu}_m$ to be valid parameters for the reconstruction of a valid empirical model. We state details on the recovery procedure in Appendix C.8.1.

Once a valid empirical model $\hat{\mathcal{B}}$ is obtained, the remaining step is to call the planning oracle that gives an $\epsilon$-approximate optimal policy for $\hat{\mathcal{B}}$. We conclude this section with an end-to-end guarantee.

**Theorem 3.8** *For any LMAB instance with $M$ latent contexts, with probability at least $1 - \eta$, Algorithm 1 returns an $\epsilon$-optimal policy given total number of episodes of at most*

$$\text{poly}(H, M, Z, A, 1/\epsilon) \cdot \log(AZ/\eta) + \text{poly}(H, Z, M)^{2M-1} \cdot \log(M/\eta)/\epsilon^{4M-2}, \quad \text{if } H \geq 2M - 1,$$

$$\text{poly}(H, w_{\min}^{-1}, Z, A, 1/\epsilon) \cdot \log(AZ/\eta) + H^2 \log(M/\eta) \cdot \left(2MZ^2\right)^{H}/\epsilon^2, \quad \text{otherwise.}$$

Note that the dependency on $A$ is polynomial. This polynomial term is needed to control the subspace estimation error. The exponential term is derived from the closeness in moments as discussed earlier.

**Remark 3.9 (Note on $w_{\min}$)** *Ideally, mixing weights are well balanced such that $w_{\min} = \Omega(1/M)$. However, in general we may have arbitrarily small $w_{\min}$. Note that when $H \geq 2M - 1$, the upper bound does not depend on $w_{\min}$. However when $H < 2M - 1$ with arbitrarily small $w_{\min}$, effectively we have $w_{\min} \propto \epsilon/(Z\sqrt{M})^H$ as can be seen in equation (5). This is mainly because in the parameter unidentifiable regime, we give guarantees from moment-closeness, and thus the subspace has to be very accurate to approximate higher-order moments well. Nevertheless, we are not aware whether this is tight, and believe this to be an interesting question for future research.*

**Continuous Rewards – Gaussian Case.** So far we have focused on rewards with finite support $Z = O(1)$. However, some steps in the algorithm cannot be straightforwardly extended to continuous reward distributions. In Appendix D, we show a similar upper bound that is at most

$$\text{poly}(H, M, A, 1/\epsilon) \cdot \log(A/\eta) + \text{poly}\left(H, M, \log(1/(\eta\epsilon))\right)^{\min(H,M)} / \epsilon^{\min(2H+2, 4M-2)},$$

assuming the rewards are Gaussian with unknown mean (see Theorem D.2 for the exact upper bound).

## 4 Maximum Likelihood Implementation

In the previous section, we derived a procedure that learns a near-optimal policy of an LMAB instance. However, our procedure relies on matching higher-order moments, which, in general, is not suitable for practical implementation. In this section, we describe a maximum likelihood (MLE) method, motivated by our previous results. Importantly, we can use the Expectation-Maximization (EM) [11] heuristic to find an approximate solution to the MLE optimization problem.

We can start from the set of core action-value pairs $\{(a_j, z_j)\}_{j=1}^n$ given by Corollary 3.2. Now for every time step $t$, we choose $i_t$ randomly from a uniform distribution over $[n]$, play action $a_{i_t}$, and observe $b_t := \mathbb{1}\{r_t = z_{i_t}\}$. After repeating this for $N$ episodes, we formulate the log-likelihood function with parameterization $\theta = \{(w_m, \nu_m)\}_{m=1}^M$ as follows:

$$l_N(\theta) := \frac{1}{N} \sum_{k=1}^N \log \left( \sum_{m=1}^M w_m \Pi_{t=1}^H (b_t \nu_m(i_t) + (1 - b_t)(1 - \nu_m(i_t))) \right) \tag{8}$$

To prevent the confusion with searching parameters, we use $q^*$ to denote any quantity $q$ constructed with ground truth parameters, *e.g.,* $\theta^* := \{(w_m^*, \nu_m^*)\}_{m=1}^M$. Let $\theta_N$ be the maximum likelihood estimator, *i.e.,* $\theta_N = \arg\max_{\theta \in \Theta} l_N(\theta)$ in some valid parameter set $\Theta$.

We can recover the sample-complexity guaranteed by moment-matching methods studied in the previous section. Specifically, we show that the maximum likelihood estimator with sufficiently many samples have nearly matching moments:

**Lemma 4.1** *Consider the maximum likelihood estimator $\theta_N = \{(\hat{w}_m, \hat{\nu}_m)\}_{m=1}^M$ with $N$ episodes for large enough $N$. If $N \geq C \cdot n^{\min(2H+1, 4M-1)} \log(N/\eta)/\delta^2$ for some sufficiently large constant $C > 0$, then with probability at least $1 - \eta$,*

$$\|T_l^* - \hat{T}_l\|_\infty \leq \delta, \qquad \forall l \in [\min(H, 2M - 1)]. \tag{9}$$

Therefore, by setting the same accuracy parameter $\delta$ used in Algorithm 1, we can obtain the required $\theta_N = \{(\hat{w}_m, \hat{\nu}_m)\}_{m=1}^M$ for matching moments. We only replace the moment-matching step (Step 2 in Algorithm 1) with solving the MLE optimization problem (8).

**Implementation Details** While the log-likelihood formulation (8) is still non-convex and thus intractable, we can rely on powerful heuristics: we initialize the parameters with clustering methods [2] or spectral methods [1, 7], and run the EM algorithm to improve the accuracy [11] (regardless of conditions or guarantees). Therefore, while the sample-complexity upper bound for both methods remains the same, we benefit from the log-likelihood formulation despite the non-convexity.

Specifically, we follow the same steps in Algorithm 1. We first find the core action-event pairs from second-order moments as described (**Step 1**). Then we form the maximum log-likelihood objective (8) and find an approximate MLE solution with the EM algorithm (**Step 2**). After obtaining $\theta_N$, we recover an empirical model which we use as an input to an approximate planning oracle (**Step 3**). We

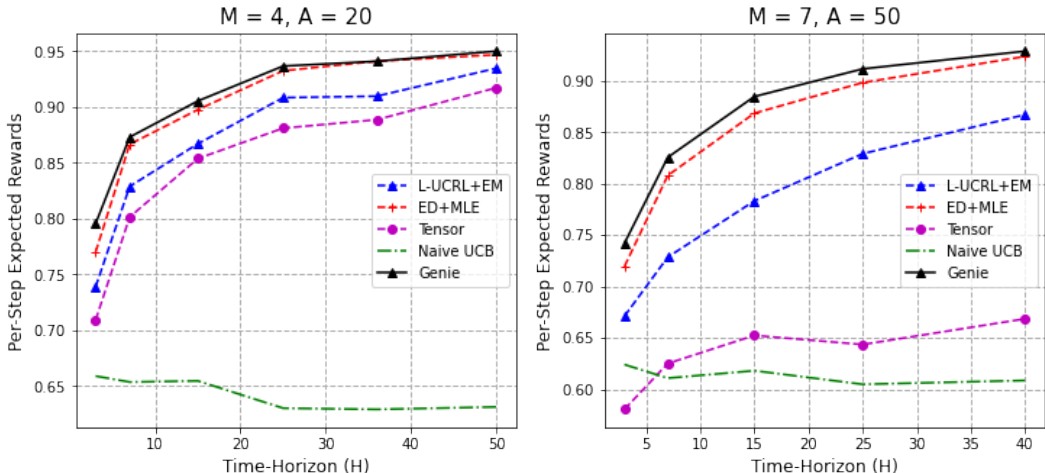

**Figure 1:** Per time-step rewards for increasing lengths of episodes with history-dependent policies returned after the exploration phase

compare our MLE-based implementation with experimental design (ED + MLE) to other baselines applicable to learning in LMABs (*e.g.,* L-UCRL + EM [24], tensor-decomposition methods [1, 7, 41], naive-UCB [6]). In our experimental results, we can observe that in practice, our method (ED+MLE) outperforms other baselines and the worst case guarantee given in Theorem 3.8.

**Provable Benefits of MLE** In Appendix E.2, we also show that MLE solutions can automatically adapt to mild separation conditions. Specifically, we show that if $H = \tilde{O}(M^2/\gamma^2)$ for some separation parameter $\gamma > 0$ (see Assumption E.1), then we can achieve the polynomial sample complexity for learning a near-optimal policy with the MLE solution $\theta_N$.

In Figure 1, we compare the averaged *per time-step* rewards obtained with returned policies from different algorithms after some exploration episodes. Note that with increasing $H$, it becomes easier to identify contexts, *i.e.*, we have bigger separations, and thus MLE based approaches can benefit from larger separations. Due to space constraints, we defer the remaining details to Appendix A.

## 5    Conclusion and Future Work

In this work, we designed an algorithm that learns a near-optimal policy for an LMAB instance that requires only $\text{poly}(A) + \text{poly}(M, H)^{\min(H, 2M-1)}$ number of samples. We achieved this with the experimental design and method-of-moments, and the maximum likelihood estimation which is more suitable in practice. We discuss a few limitations of this work and future directions below.

**Lower Bounds** The question of a lower bound on the sample complexity of the LMAB problem remains unresolved. For LMDP, an MDP extension of LMAB, a lower bound of $\Omega(A^M)$ is known due to [24]. While we conjecture that some exponential dependence in $M$ is unavoidable, characterizing the minimax dependence of the exponent is left as an interesting future research question.

**Latent MDPs** We believe that the moment-matching based approach we took in this work offers a promising way for designing RL algorithms in the presence of latent contexts. Specifically, we believe these techniques can be used to design algorithm that finds a near-optimal policy of LMDPs [24] or reward-mixing MDPs [23] with $M = O(1)$ and without further separation assumptions.

**Linear Bandits / Continuous Rewards** Our work has focused on the tabular setting, where all arms are independent. It will be an interesting future work to consider the same objective in linear bandit settings. Also, while we only considered Gaussian rewards, we believe our approach can also be extended to a broader class of parametric distributions. Investigating more general classes of rewards is an important future research direction from practical perspective.

**Acknowledgement**

This research was partially funded by the NSF IFML Institute (NSF 2019844), the NSF AI-EDGE Institute (NSF 2112471), and NSF 1704778, and was also supported by the Israel Science Foundation (grant No. 2199/20).

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
