

**Figure 2:** Left: Performance with increasing $M$, Right: Performance with increasing exploration episodes

## Appendix A   Experiments

In this section, we demonstrate our maximum likelihood estimation (8) on synthetic examples. We generate random LMAB instances with Bernoulli rewards, where the mean-reward vectors $\{\mu_m\}_{m=1}^M$ lie in a subspace of dimension roughly 4, *i.e.,* rank $\left(\sum_{m=1}^M w_m \mu_m \mu_m^\top\right) \approx 4$. We compare our MLE with experimental design method (8) (ED+MLE) to three benchmarks:

1. Naive UCB [4] without considering contexts (thus a returned policy is stationary).
2. Tensor-decomposition methods [1, 41].
3. L-UCRL with spectral initialization [24].

Even when the theoretical conditions required for the success of tensor-decomposition or spectral methods do not, in practice, standard tensor-decomposition technique by [1] serves as good initialization for the EM or L-UCRL algorithms. After estimating the LMAB $\hat{\mathcal{B}}$, we compute a heuristic policy using Q-MDP [28] for $\hat{\mathcal{B}}$. We refer to the computed policy by Q-MDP [28] using the true LMAB model $\mathcal{B}$ as the *genie policy*.

In the first experiment, we compare the performance of the aforementioned four alternatives. We draw a random LMAB instance with $M = 4, A = 20$ using $N = 5 \cdot 10^4$ sampled episodes, and $M = 7, A = 50$ using $N = 100000$ episodes (Figure 1). We compare the averaged *per time-step* rewards obtained with each policy with increasing length of episodes $H$.

When $M = 4$, all methods (except naive UCB) exhibit the same pattern of improved performance as the algorithms access more data. As we generated instances to satisfy the full-rank condition, *i.e.,* rank $\left(\sum_{m=1}^M w_m \mu_m \mu_m^\top\right) \approx 4$, even pure tensor-decomposition method works well in practice in this setting. However, when $M = 7$, pure tensor-decomposition method significantly under-performs L-UCRL or ED+MLE. This demonstrates that for LMAB instances with rank degeneracy, additional iteration steps with EM are necessary to get a good solution. Furthermore, the performance of ED+MLE and of L-UCRL does not significantly drop in $M = 7$. This demonstrated that practically it works much better than what can be guaranteed in theory in the worst case. We conjecture that this is because the EM iteration converges to the MLE (even when converging to local optimums), and MLE solutions in general show much better performance under some mild conditions. (*e.g.,* with Assumption E.1, or if random perturbations are applied to the underlying LMAB model [8]).

In our second experiment, we test the performance of the different methods while scanning different control parameters. We first fix $A = 50$, $H = 7$, and change the number of contexts (left of Figure 2). Since we keep generating instances satisfying rank $\left(\sum_{m=1}^M w_m \mu_m \mu_m^\top\right) \approx 4$, tensor-based method performs significantly worse as $M$ increases. We then fix $M = 5$ and observe the performance gain by

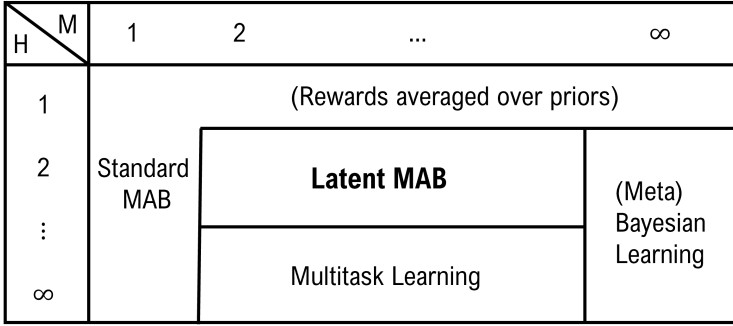

**Figure 3:** Nature of the problem depends on the length of episodes ($H$) and number of different contexts ($M$)

increasing the sample episodes (right of Figure 2). We see that the performance of both L-UCRL+EM and our method (ED+MLE) approaches to that of genie as we increase $N$.

From both experiments, we conclude that, in practice, MLE based methods perform much better than the worst case guarantee shown via method of moments. We believe it will be an interesting future direction whether we can derive much better guarantees for MLE based solutions.

## Appendix B    Additional Preliminaries

### B.1    Additional Related Work

**Comparison of Regimes**    Nature of learning a near-optimal policy in LMABs might change depending on the length of episodes $H$ and the number of contexts $M$ (recall Figure 3). For instance, when $H = 1$ or $M = 1$, the problem of learning in LMAB is essentially equivalent to the classical Multi-Armed Bandits problem that has been extensively studied in literature (*e.g.,* see [26] and references therein). In another well-studied literature of Bayesian learning, each episode is assumed to have a structured prior over contexts ($M \to \infty$, *e.g.,* all expected rewards of arms are independently sampled from beta conjugate priors). Depending on the length of time-horizon $H$, the problem can be solved with Thompson Sampling (TS) [38] $H \to \infty$, or the problem is reduced to learning the prior [21]. When the time-horizon is sufficiently long but finite *e.g.,* if $H = \Omega(A/\gamma^2)$ for some separation parameters between a finite number of contexts $M < \infty$, then it is possible cluster observations from every episode. This setting has been studied in the literature of multitask RL which we describe below.

In this work, we are particularly focused on the setting with a few contexts and relatively short episodes (in comparison to $A$), *i.e.,* $M = O(1)$ and $H = O(1)$, where the most natural objective is to learn a near-optimal history-dependent policy.

**Reward-Mixing MDPs**    Another closely related work is to learn an LMDP (MDP extension of LMAB) with common state transition probabilities, and thus only the reward function changes depending on latent contexts [23]. The authors in [23] have developed an efficient moment-matching based algorithm to learn a near-optimal policy without any assumptions on separations or system dynamics. However, it can only handle the $M = 2$ case with balanced mixing weights $w_1 = w_2 = 1/2$. It is currently not obvious how to extend their result to $M \geq 3$ cases without incurring $O(A^M)$ sample complexity in LMABs.

**Regime switching bandits**    LMAB may be also seen as a special type of adversarial or non-stationary bandits (*e.g.,* [5, 3, 13, 36]) with time steps being specified for when the underlying reward distributions (*i.e.,* latent contexts) may change. The standard objective in non-stationary bandits is to find the best stationary policy in hindsight with unlimited possible contexts. Recently, [41] considered a non-stationary bandit with a finite number of contexts $M = O(1)$ and the objective of finding the optimal *history-dependent* policy. Their setting and goal subsume our goal of learning the optimal policy in LMABs; however, results in [41] require linear independence between reward probability vectors, and thus their setting essentially falls into the category of tractable POMDPs with full-rank observations.

**Multitask RL with explicit clustering**    We can cluster observations from each episode if we are given sufficiently long time horizons $H = \tilde{\Omega}(A/\gamma^2)$ in each episode [9, 14, 16]. Here, $\gamma > 0$ is the amount of 'separation' between contexts such that for all $m \neq m'$, $\max_{a \in \mathcal{A}} \|\mu_m(a, \cdot) - \mu_{m'}(a, \cdot)\|_2 \geq \gamma$ where $\mu_m$ is a mean-reward vector of actions in the $m^{th}$ context. We focus on significantly more general cases where there is no obvious way of clustering observations, *e.g.,* when $H \ll A$ or $\mu_m$ can be arbitrarily close to some other $\mu_{m'}$ for $m \neq m'$. If we are given a similar separation condition, we also show that a polynomial sample complexity is achievable as long as $H = \tilde{O}(M^2/\gamma^2)$ (see also section E.2). Note that this could still be in $H \ll A$ regime with large number of actions $A$.

**Miscellaneous**    There are other modeling approaches where multiple episodes proceed in parallel without limits on the time-horizon [31, 14, 10, 17, 25]. In such problems, the goal is to quickly adapt policies for individual episodes assuming certain similarities between tasks. In contrast, in episodic LMAB settings, we assume that every episode starts in a sequential order, and our goal is to learn an optimal history-dependent policy that can maximize rewards for a single episode with limited time horizon.

## B.2   Experimental Design

We now give a high-level overview on experimental design techniques used in this work. Suppose we are given a matrix $\Phi \in \mathbb{R}^{d \times k}$ where $d \gg k$. Define a distribution over the rows of $\Phi$, $\rho \in \Delta^d$ an element in the $d$-dimensional simplex. We want to select a small subset of rows of $\Phi$ which minimizes $g(\rho)$ defined below:

$$G(\rho) = \sum_{i \in [d]} \rho(i)\Phi_{i,:}\Phi_{i,:}^\top, \qquad g(\rho) = \max_{i \in [d]} \|\Phi_{i,:}\|_{G(\rho)^{-1}}^2, \tag{10}$$

where $\Phi_{i,:} \in \mathbb{R}^k$ be the $i^{\text{th}}$ row of $\Phi$ and $G(\rho) \in \mathbb{R}^{k \times k}$. To achieve this task we use results from the experimental design literature. The following theorem shows the existence of $\rho$ that minimizes $g(\rho)$ with *a small support* over the row indices of $\Phi$:

**Theorem B.1 (Theorem 4.4 in [27])**  *There exists a probability distribution $\rho$ such that $g(\rho) \leq 2k$ and $|supp(\rho)| \leq 4k \log \log k + 16$. Furthermore, we can compute such $\rho$ in time $\tilde{O}(dk^2)$.*

As noted in [27], Theorem B.1 can be obtained from results of Chapter 3 in [39]. Using this fundamental theorem, [27] showed the following proposition:

**Proposition B.2 (Proposition 4.5 in [27])**  *Let $\rho$ be a distribution over the rows of $\Phi$ that satisfies the condition of Theorem B.1. Suppose a vector $\mu \in \mathbb{R}^d$ can be represented as a sum $\mu = v + \Delta$ where $v \in V$, $\|\Delta\|_\infty \leq \epsilon_0$. Let $\eta$ be any small noise with $\eta \in [-\epsilon_1, \epsilon_1]^d$. Then $\|\Phi\hat{\theta} - \mu\|_\infty \leq \epsilon_0 + (\epsilon_0 + \epsilon_1)\sqrt{2k}$ where*

$$\hat{\theta} = G(\rho)^{-1} \sum_{i \in [d]} \rho(i)(\mu(i) + \eta(i))\Phi_{i,:}. \tag{11}$$

Crucially, we use Proposition B.2 to reduce the sample complexity in $A$.

## B.3   Wasserstein Distance

We now give a brief overview on the Wasserstein distance and its applications in latent mixture models. Wasserstein distance is a convenient error metric to measure the parameter distance between two latent models $\{(w_m, \nu_m)\}_{m=1}^M$ and $\{(\hat{w}_m, \hat{\nu}_m)\}_{m=1}^M$, where $w_m, \hat{w}_m$ are mixing probabilities and $\nu_m, \hat{\nu}_m$ are some parameters for individual contexts.

Wasserstein distance is defined as follows. Let $\nu$ be a finite-support distribution over $\{\nu_m\}_{m=1}^M$ with probabilities $\{w_m\}_{m=1}^M$, *i.e.,* $\nu = \sum_{m=1}^M w_m \delta_{\nu_m}$ where $\delta_v$ is a Direc-delta distribution with a single mass on $v \in \mathbb{R}^n$. Similarly with parameters $\{(\hat{w}_m, \hat{\nu}_m)\}_{m=1}^M$, define an atomic distribution

$\hat{\nu} = \sum_m \hat{w}_m \delta_{\hat{\nu}_m}$. We define a Wasserstein distance between $\nu$ and $\hat{\nu}$ with respect to $l_\infty$ norm as the following:

$$W(\nu, \hat{\nu}) := \inf_\Gamma \mathbb{E}_{(m,m')\sim\Gamma} \left[\|\nu_m - \hat{\nu}_{m'}\|_\infty\right] = \inf_\Gamma \sum_{(m,m')\in[M]^2} \Gamma(m, m') \cdot \|\nu_m - \hat{\nu}_{m'}\|_\infty,$$

where the infimum is taken over all couplings over joint distributions $\nu$ and $\hat{\nu}$ which are marginally distributed as $\nu$ and $\hat{\nu}$ respectively (identical to (3)), *i.e.*,

$$\Gamma(m, m') \in \mathbb{R}_+^{M\times M} : \sum_{m'=1}^{M} \Gamma(m, m') = w_m, \sum_{m=1}^{M} \Gamma(m, m') = \hat{w}_m.$$

One nice property of Wasserstein metric is that the distance measure is invariant to permutation of individual components, and flexible with arbitrarily small mixing probabilities or close parameters for different contexts [40, 12].

# Appendix C Proofs for Section 3

## C.1 Proof of Lemma 3.1

Let $\{\beta_j\}_{j=1}^{k} \subseteq \mathbb{R}^d$ be orthonormal basis of a $k$-dimensional subspace $\mathbf{U}$. Let $\Phi \in \mathbb{R}^{d\times k}$ be a matrix of form $[\beta_1 \quad \beta_2 \quad ... \quad \beta_k]$, *i.e.*, the $j^{\text{th}}$ column of $\Phi$ is $\beta_j$. We need to show the existence of a small set of core coordinates, from which we can reconstruct $\mu \in \mathbf{U}$.

Proposition B.2 implies that we can find a set of coordinates $\{i_j\}_{j=1}^{n}$ with cardinality at most $n = 4d \log \log d + 16$, such that if $\mu \in \mathbf{U}$ and we can access the vector $[\mu(i_1), \ldots, \mu(i_n)]$ up to accuracy $\epsilon_1$, then we can reconstruct $\mu$ up to $\epsilon_1\sqrt{2k}$ error. We can also infer that for any $\mu \in \mathbf{U}$, we have $\Phi\theta = \mu$ where $\theta$ is given by $G(\rho)^{-1} \sum_{i\in[d]} \rho(i)\mu(i)a_i$.

To conclude Lemma 3.1, suppose we find $\hat{\mu} \in \mathbf{U}$ such that $|\mu(i_j) - \hat{\mu}(i_j)| \leq \epsilon_0$ for all $j \in [n]$. Let $\eta$ be such that $\eta(i) = \mu(i) - \hat{\mu}(i)$ if $i \in \{i_j\}_{j=1}^{n}$, and $\eta(i) = 0$ otherwise. Applying Proposition B.2 with $\Delta = 0$ and $\eta$ be as defined above, with $\hat{\mu} = \Phi\hat{\theta}$, we have $\|\hat{\mu} - \mu\|_\infty \leq \epsilon_0\sqrt{2k}$.

## C.2 Proof of Corollary 3.2

Let $\{\beta_m\}_{m=1}^{M} \subseteq \mathbb{R}^{AZ}$ be orthonormal basis of a $M$-dimensional subspace that includes $\mathbf{U} = span(\{\mu_m\}_{m=1}^{M})$. Then by Lemma 3.1, we have

$$\max_{a\in\mathcal{A}} \|\mu_m(a, \cdot) - \hat{\mu}_m(a, \cdot)\|_1 = \max_{a\in\mathcal{A}} \sum_{z\in\mathcal{Z}} |\mu_m(a, z) - \hat{\mu}_m(a, z)|$$
$$\leq 2Z\sqrt{2M} \max_{j\in[n]} |\mu_m(a_j, z_j) - \hat{\mu}_m(a_j, z_j)|,$$

with a set of core action-value pairs $\{(a_j, z_j)\}_{j=1}^{n}$. Plugging $v_m(j) = \mu_m(a_j, z_j)$ and $\hat{\nu}_m(j) = \hat{\mu}_m(a_j, z_j)$ gives Corollary 3.2.

**Remark C.1 (Eliminating factor $Z$ for $H \geq M$)** *Instead of core action-value pairs, we can also define core action-event pairs to save a factor of $Z$. For instance, define a basis $\{\tilde{\beta}_m\}_{m=1}^{M}$ in a lifted space $\mathbb{R}^{A\times 2^Z}$ defined as the following:*

$$\tilde{\beta}_m(a, S) := \sum_{z\in S} \beta_m(a, z), \qquad \forall a \in \mathcal{A}, S \subseteq \mathcal{Z}.$$

*For each $\mu_m$, define $\phi_m$:*

$$\phi_m(a, S) := \sum_{z\in S} \mu_m(a, z), \qquad \forall a \in \mathcal{A}, S \subseteq \mathcal{Z}. \tag{12}$$

By definition, $span(\{\phi_m\}_{m=1}^M) \subseteq span(\{\tilde{\beta}_m\}_{m=1}^M)$. Since $\frac{1}{2}\|\mu_m(a, \cdot) - \hat{\mu}_m(a, \cdot)\|_1 = \max_{S \subseteq \mathcal{Z}} |\phi_m(a, S) - \hat{\phi}_m(a, S)|$, it follows from Lemma 3.1 that we can find a set of core action-events $\{(a_j, \mathcal{Z}_j)\}_{j=1}^n$ of size at most $n = 4M \log \log M + 16$ such that for any $\phi_m, \hat{\phi}_m \in V$, we have

$$\|\phi_m - \hat{\phi}_m\|_\infty \leq \sqrt{2M} \cdot \max_{j \in [n]} |\phi_m(a_j, \mathcal{Z}_j) - \hat{\phi}_m(a_j, \mathcal{Z}_j)|.$$

*This approach enables to approximate reward distributions in $l_1$ statistical distance, and thus we can save a factor of $Z$ in subsequent analysis. However, it comes with exponentially more expensive (in $Z$) computations for finding the core set.*

## C.3 Proof of Lemma 3.3

To prove this result, we use the recent results on converting closeness in higher-order moments to closeness in Wasserstein distances for atomic distributions [40, 12]. A key result in [40] states a connection between moments and Wasserstein distance in one-dimensional case:

**Theorem C.2 (Proposition 1 in [40])** *If $n = 1$ and $\left|\sum_{m=1}^M w_m \nu_m^d - \sum_{m=1}^M \hat{w}_m \hat{\nu}_m^d\right| < \delta$ for all $d = 1, 2, ..., 2M - 1$, then*

$$W(\gamma, \hat{\gamma}) \leq O\left(M\delta^{1/(2M-1)}\right).$$

Theorem C.2, which holds for 1-dimensional mixtures–can be generalized to the high-dimensional case, as the following theorem shows.

**Theorem C.3** *For $n \geq 2$, if $\left\|\sum_{m=1}^M w_m \nu_m^d - \sum_{m=1}^M \hat{w}_m \hat{\nu}_m^d\right\|_\infty < \delta$ for all $d = 1, 2, ..., 2M - 1$, then*

$$W(\gamma, \hat{\gamma}) \leq O\left(M^3 n \cdot \delta^{1/(2M-1)}\right).$$

*Proof.* The idea follows the proof of Lemma 3.1 in [12]. Suppose a standard Gaussian random variable $\theta \sim \mathcal{N}(0, I)$ in $\mathbb{R}^n$. For any $x \in \mathbb{R}^n$, anti-concentration of Normal distribution says

$$\mathbb{P}(|\theta^\top x| \leq \tau \|x\|_2) \leq \tau,$$

for any $\tau > 0$. By union bound and the fact that $\|x\|_2 \geq \|x\|_\infty$, if we define $\mathcal{X} = \{v - \hat{v} | v \in \{\nu_m\}_{m=1}^M, \hat{v} \in \{\hat{\nu}_m\}_{m=1}^M\}$, then we have

$$\mathbb{P}(|\theta^\top x| \leq \tau \|x\|_\infty) \leq M^2 \tau, \qquad \forall x \in \mathcal{X}.$$

Also with high probability, we have $\mathbb{P}(\|\theta\|_1 \geq 2n) \leq \frac{n\mathbb{E}_{t \sim \mathcal{N}(0,1)}[|t|]}{2n} < 1/2$ by Markov inequality. Thus,

$$\mathbb{P}\left(\frac{\theta^\top x}{\|\theta\|_1} > \frac{\tau}{2n}\|x\|_\infty\right) > 1/2 - M^2\tau.$$

By setting $\tau = M^2/2$, this probabilistic argument implies that there exists $\theta \in \mathbb{R}^n$ with a unit $L_1$ norm, $\|\theta\|_1 = 1$, such that for all $x \in \mathcal{X}$, we have $\|x\|_\infty \leq 4M^2 n |\theta^\top x|$. Using this $\theta$, we have

$$W(\gamma, \hat{\gamma}) = \inf_\Gamma \{\mathbb{E}_{(m,m') \sim \Gamma}[\|\nu_m - \hat{\nu}_{m'}\|_\infty]\} \leq 4M^2 n \cdot \inf_\Gamma \{\mathbb{E}_{(m,m') \sim \Gamma}[|\theta^\top(\nu_m - \hat{\nu}_{m'})|]\}. \quad (13)$$

On the other hand, consider 1-dimensional $M$-atomic distributions $\gamma_\theta := \sum_{m=1}^M w_m \delta_{\theta^\top \nu_m}$ and $\hat{\gamma}_\theta := \sum_{m=1}^M \hat{w}_m \delta_{\theta^\top \hat{\nu}_m}$. Then by Cauchy-Schwartz inequality

$$\left|\sum_{m=1}^M w_m (\theta^\top \nu_m)^d - \sum_{m=1}^M \hat{w}_m (\theta^\top \hat{\nu}_m)^d\right| \leq \|\theta\|_1^d \left\|\sum_{m=1}^M w_m \nu_m^{\otimes d} - \sum_{m=1}^M \hat{w}_m \hat{\nu}_m^{\otimes d}\right\|_\infty \leq \delta,$$

for all $d = 1, 2, ..., 2M - 1$. Since $\gamma_\theta, \hat{\gamma}_\theta$ are 1-dimensional atomic distributions, by Theorem C.2,

$$W(\gamma_\theta, \hat{\gamma}_\theta) := \inf_\Gamma \{\mathbb{E}_{(m,m') \sim \Gamma}[|(\theta^\top \nu_m) - (\theta^\top \hat{\nu}_{m'})|]\} \leq O\left(M\delta^{1/(2M-1)}\right).$$

Note that any coupling over $\nu_\theta, \hat{\nu}_\theta$ can be converted to a coupling over $\nu, \hat{\nu}$ and vice versa. Plugging the above result into (13), we have the theorem. □

With the above theorems, if we set $\delta < (\epsilon/(M^3 n))^{2M-1}$, we have $W(\gamma, \hat{\gamma}) \leq O(\epsilon)$.

### C.4 Proof of Proposition 3.4

First note that the difference in expected values of a policy is bounded by total variation distance between trajectory distribution:

$$|V(\pi) - \hat{V}(\pi)|$$
$$\leq H \cdot \sum_{\substack{r_{1:H} \in \mathcal{R}^H, \\ a_{1:H} \in \mathcal{A}^H}} |\mathbb{P}^\pi(r_{1:H}, a_{1:H}) - \hat{\mathbb{P}}^\pi(r_{1:H}, a_{1:H})|$$
$$\leq H \sum_{r_{1:H}, a_{1:H}} \left| \sum_{m=1}^M w_m \Pi_{t=1}^H \mu_m(a_t, r_t) - \sum_{m=1}^M \hat{w}_m \Pi_{t=1}^H \hat{\mu}_m(a_t, r_t) \right| \cdot \Pi_{t=1}^H \pi(a_t | a_{1:t-1}, r_{1:t-1}).$$

Now we only need to focusing on total variation distance bound. Fron this point, when we sum over sequences, we sum over all possible sequences if not specified. With a slight abuse in notation, we use a compact notation for probability of action sequences:

$$\pi(a_{1:t} | r_{1:t-1}) := \Pi_{t'=1}^t \pi(a_{t'} | r_{1:t'-1}, a_{1:t'-1}). \tag{14}$$

First, suppose any coupling $\Gamma$ between contexts $m$ and $m'$ in two systems, and we can write

$$\sum_{r_{1:H}, a_{1:H}} \left| \sum_{m=1}^M w_m \Pi_{t=1}^H \mu_m(a_t, r_t) - \sum_{m=1}^M \hat{w}_m \Pi_{t=1}^H \hat{\mu}_m(a_t, r_t) \right| \cdot \pi(a_{1:H} | r_{1:H-1})$$
$$\leq \sum_{r_{1:H}, a_{1:H}} \pi(a_{1:H} | r_{1:H-1}) \sum_{(m,m')} \Gamma(m, m') \left| \Pi_{t=1}^H \mu_m(a_t, r_t) - \Pi_{t=1}^H \hat{\mu}_{m'}(a_t, r_t) \right|,$$

where the inequality holds by the triangle inequality. Now we can proceed as

$$\sum_{(m,m')} \Gamma(m, m') \sum_{r_{1:H}, a_{1:H}} \pi(a_{1:H} | r_{1:H-1}) \left| \Pi_{t=1}^H \mu_m(a_t, r_t) - \Pi_{t=1}^H \hat{\mu}_{m'}(a_t, r_t) \right|$$
$$\leq \sum_{(m,m')} \Gamma(m, m') \cdot \left( \sum_{r_{1:H-1}, a_{1:H}} \pi(a_{1:H} | r_{1:H-1}) \cdot \Pi_{t=1}^{H-1} \mu_m(a_t, r_t) \sum_{r_H} |\mu_m(r_H; a_H) - \hat{\mu}_{m'}(a_H, r_H)| \right.$$
$$\left. + \sum_{r_{1:H-1}, a_{1:H}} \pi(a_{1:H} | r_{1:H-1}) \cdot \left| \Pi_{t=1}^{H-1} \mu_m(a_t, r_t) - \Pi_{t=1}^{H-1} \hat{\mu}_{m'}(a_t, r_t) \right| \cdot \sum_{r_H} |\hat{\mu}_{m'}(a_H, r_H)| \right),$$

where we used triangle inequality at the $t = H$ time step. Note that $\sum_{r_H} |\mu_m(a_H, r_H) - \hat{\mu}_m(a_H, r_H)| = \|\mu(a_H, \cdot) - \hat{\mu}_m(a_H, \cdot)\|_1$. Thus, we can bound the first term by

$$\sum_{r_{1:H-1}, a_{1:H}} \pi(a_{1:H} | r_{1:H-1}) \cdot \Pi_{t=1}^{H-1} \mu_m(a_t, r_t) \sum_{r_H} |\mu_m(a_H, r_H) - \hat{\mu}_{m'}(a_H, r_H)|$$
$$\leq \max_{a \in \mathcal{A}} \|\mu_m(a, \cdot) - \hat{\mu}_{m'}(a, \cdot)\|_1.$$

where we summed over all probabilities over sequences where we used the fact that

$$\sum_{r_{1:H-1}, a_{1:H}} \pi(a_{1:H} | r_{1:H-1}) \cdot \Pi_{t=1}^{H-1} \mu_m(a_t, r_t) = \sum_{r_{1:H-1}, a_{1:H}} \mathbb{P}_m^\pi(r_{1:H-1}, a_{1:H}) = 1.$$

For the second term, we observe that

$$\sum_{r_{1:H-1}, a_{1:H}} \pi(a_{1:H} | r_{1:H-1}) \left| \Pi_{t=1}^{H-1} \mu_m(a_t, r_t) - \Pi_{t=1}^{H-1} \hat{\mu}_{m'}(a_t, r_t) \right| \sum_{r_H} |\hat{\mu}_{m'}(a_H, r_H)|$$
$$= \sum_{r_{1:H-1}, a_{1:H-1}} \pi(a_{1:H-1} | r_{1:H-2}) \left| \Pi_{t=1}^{H-1} \mu_m(a_t, r_t) - \Pi_{t=1}^{H-1} \hat{\mu}_{m'}(a_t, r_t) \right|.$$

since $\sum_{r_H} \hat{\mu}_{m'}(a, r_H) = 1$ for any $a$. From here, we can apply the same decomposition which we used at $t = H$ level, and apply the same argument recursively until $t = 1$. That gives for any coupling $\Gamma$,

$$\sum_{r_{1:H}, a_{1:H}} \left| \sum_{m=1}^M w_m \Pi_{t=1}^H \mu_m(a_t, r_t) - \sum_{m=1}^M \hat{w}_m \Pi_{t=1}^H \hat{\mu}_m(a_t, r_t) \right| \cdot \pi(a_{1:H} | r_{1:H-1})$$

$$\leq H \cdot \sum_{(m,m')} \Gamma(m,m') \max_{a \in \mathcal{A}} \|\mu_m(a,\cdot) - \hat{\mu}_{m'}(a,\cdot)\|_1.$$

Since the inequality holds for all valid couplings $\Gamma(m,m')$ such that

$$\sum_m \Gamma(m,m') = \hat{w}_{m'}, \text{ and, } \sum_{m'} \Gamma(m,m') = w_m,$$

we can take the infimum over $\Gamma$, and conclude that

$$|V(\pi) - \hat{V}(\pi)| \leq H^2 \cdot \inf_\Gamma \sum_{(m,m')} \Gamma(m,m') \max_{a \in \mathcal{A}} \|\mu_m(a,\cdot) - \hat{\mu}_{m'}(a,\cdot)\|_1.$$

## C.5  Proof of Lemma 3.5

In order to bound difference between expected long-term rewards for a fixed history-dependent policy $\pi$, it is sufficient to bound the difference in distributions of observations following $\pi$. We first explicitly write down total variation distance between observations from $\{(w_m, \mu_m)\}_{m=1}^M$ and $\{(\hat{w}_m, \hat{\mu}_m)\}_{m=1}^M$:

$$\sum_{r_{1:H}, a_{1:H}} |\mathbb{P}^\pi(r_{1:H}, a_{1:H}) - \hat{\mathbb{P}}^\pi(r_{1:H}, a_{1:H})|$$

$$= \sum_{r_{1:H}, a_{1:H}} \left| \sum_{m=1}^M w_m \Pi_{t=1}^H \mu_m(a_t, r_t) - \sum_{m=1}^M \hat{w}_m \Pi_{t=1}^H \hat{\mu}_m(a_t, r_t) \right| \pi(a_{1:H}|r_{1:H-1})$$

$$\leq \delta \sum_{r_{1:H}, a_{1:H}} \Pi_{t=1}^H \pi(a_t|a_{1:t-1}, r_{1:t-1}) = Z^H \cdot \delta,$$

where $\pi(a_{1:H}|r_{1:H-1})$ is as defined in (14), in the first inequality we used the condition $\|\sum_{m=1}^M w_m \mu_m^{\otimes H} - \sum_{m=1}^M \hat{w}_m \hat{\mu}_m^{\otimes H}\|_\infty \leq \delta$, and the last inequality follows from that:

$$\sum_{r_{1:H}, a_{1:H}} \pi(a_{1:H}|r_{1:H-1}) = \sum_{r_{1:H}} \sum_{a_{1:H}} \Pi_{t=1}^H \pi(a_t|a_{1:t-1}, r_{1:t-1})$$

$$= \sum_{r_{1:H} \in \mathcal{R}^H} \sum_{a_{1:H-1} \in \mathcal{A}^{H-1}} \Pi_{t=1}^{H-1} \pi(a_t|a_{1:t-1}, r_{1:t-1}) \sum_{a_H \in \mathcal{A}} \pi(a_H|a_{1:H-1}, r_{1:H-1})$$

$$= \sum_{r_{1:H} \in \mathcal{R}^H} \sum_{a_{1:H-1} \in \mathcal{A}^{H-1}} \Pi_{t=1}^{H-1} \pi(a_t|a_{1:t-1}, r_{1:t-1}) = ... = \sum_{r_{1:H} \in \mathcal{R}^H} 1 = Z^H.$$

Now the difference in expected rewards follows as

$$|f(\pi) - \hat{f}(\pi)| \leq H \cdot \sum_{r_{1:H}, a_{1:H}} |\mathbb{P}^\pi(r_{1:H}, a_{1:H}) - \hat{\mathbb{P}}^\pi(r_{1:H}, a_{1:H})| \leq H Z^H \cdot \delta.$$

## C.6  Proof of Proposition 3.6

To show this result, we first express each coordinate of $\mu_m$ in terms of $\nu_m$. That is, for all $(a,z) \in \mathcal{A} \times \mathcal{Z}$, using Proposition B.2, by setting $\eta = 0$ we can express $\mu_m$ as

$$\mu_m = T\nu_m,$$

for some linear mapping $T \in \mathbb{R}^{AZ \times n}$ (see Section C.8.1 for details on $T$). Furthermore, from the conclusion of Proposition B.2 which implies the robustness of $\mu_m$ against the perturbation of $\nu_m$ in $l_\infty$ norm, we can infer that every row of $T$ has $l_1$ norm bounded by $\sqrt{2M}$, i.e., $\|T_{(a,z),:}\|_1 \leq \sqrt{2M}$ for all $(a,z) \in \mathcal{A} \times \mathcal{Z}$.

To bound $\|\sum_{m=1}^M w_m \mu_m^{\otimes l} - \sum_{m=1}^M \hat{w}_m \hat{\mu}_m^{\otimes l}\|_\infty$, we only need to check one entry of tensors at any position $((a_1, z_1), ..., (a_l, z_l))$ and all other entries are bounded in a similar fashion. We first check that

$$\sum_{m=1}^M w_m \cdot \Pi_{i=1}^l \mu_m(a_i, z_i) - \sum_{m=1}^M \hat{w}_m \cdot \Pi_{i=1}^l \hat{\mu}_m(a_i, z_i)$$

$$= \sum_{m=1}^{M} w_m \cdot \Pi_{i=1}^{l} \left( \sum_{j=1}^{n} \nu_m(j) T_{(a_i,z_i),j} \right) - \sum_{m=1}^{M} \hat{w}_m \cdot \Pi_{i=1}^{l} \left( \sum_{j=1}^{n} \hat{\nu}_m(j) T_{(a_i,z_i),j} \right),$$

Unfolding the product expression over $i$ for the original parameter part $\nu$,

$$\sum_{m=1}^{M} w_m \cdot \Pi_{i=1}^{l} \left( \sum_{j=1}^{n} \nu_m(j) T_{(a_i,z_i),j} \right) = \sum_{m=1}^{M} w_m \sum_{(j_1,j_2,\dots,j_l) \in [n]^l} \Pi_{k=1}^{l} \nu_m(j_k) \cdot \Pi_{k=1}^{l} T_{(a_k,z_k),j_k}$$

$$= \sum_{(j_1,j_2,\dots,j_l) \in [n]^l} \Pi_{k=1}^{l} T_{(a_k,z_k),j_k} \cdot \sum_{m=1}^{M} w_m \Pi_{k=1}^{l} \nu_m(j_k).$$

Plugging this expression, we conclude the proof:

$$\left| \sum_{m=1}^{M} w_m \cdot \Pi_{i=1}^{l} \mu_m(a_i,z_i) - \sum_{m=1}^{M} \hat{w}_m \cdot \Pi_{i=1}^{l} \hat{\mu}_m(a_i,z_i) \right|$$

$$= \left| \sum_{(j_1,j_2,\dots,j_l) \in [n]^l} \Pi_{k=1}^{l} T_{(a_k,z_k),j_k} \cdot \left( \sum_{m=1}^{M} w_m \Pi_{k=1}^{l} \nu_m(j_k) - \sum_{m=1}^{M} \hat{w}_m \Pi_{k=1}^{l} \hat{\nu}_m(j_k) \right) \right|$$

$$\leq \delta \cdot \sum_{(j_1,j_2,\dots,j_l) \in [n]^l} \left| \Pi_{k=1}^{l} T_{(a_k,z_k),j_k} \right| = \delta \cdot \Pi_{k=1}^{l} \left( \sum_{j=1}^{n} |T_{(a_k,z_k),j}| \right) \leq \delta(2M)^{l/2},$$

which is what we needed to show.

## C.7  Proof of Lemma 3.7

For each episode $k$ where $k \in [N_0]$, let the first and second actions be $a_1^k, a_2^k \sim \text{Unif}(\mathcal{A})$, and let $r_1^k, r_2^k$ be the corresponding reward feedback. We construct an empirical second-order moments $\hat{M}_2$ such that $\hat{M}_2(i,j)$ is the mean of $r_1 \cdot r_2$ when $a_1^k = a_i, a_2^k = a_j$. Specifically, we construct $\hat{M}_2$ as the following:

$$\hat{M}_2 = \frac{1}{2N_0} \sum_{k=1}^{N_0} \boldsymbol{e}_{(a_1^k,r_1^k)} \cdot \boldsymbol{e}_{(a_2^k,r_2^k)}^{\top} + \boldsymbol{e}_{(a_2^k,r_2^k)} \cdot \boldsymbol{e}_{(a_1^k,r_1^k)}^{\top}, \tag{15}$$

where $\boldsymbol{e}_{(a_t^k,r_t^k)}$ is a standard basis vector in $\mathbb{R}^{AZ}$ with 1 at position $(a_t^k, r_t^k)$. The argument follows from a rather standard concentration argument for dimensionality reduction (e.g., Lemma 3.5 in [12]). Let $\delta = \|M_2 - \hat{M}_2\|_2$. With standard measure of concentration arguments, we can show that $\|M_2 - \hat{M}_2\|_{\infty} < C \cdot \sqrt{A^2 \log(AZ/\eta)/N_0}$ with probability at least $1 - \eta$, which is translated to $\delta \leq C \cdot \sqrt{\frac{A^4 Z^2 \log(AZ/\eta)}{N_0}}$ for some universal constant $C > 0$.

Let $P_{\mathbf{U}}$ be the orthogonal projector onto the top-$M$ eigenspace of $M_2$ and $P_{\mathbf{U}}^{\perp} = I - P_{\mathbf{U}}$. We can also define similar quantities from the empirical estimate $\hat{M}_2$. Let $P_{\hat{\mathbf{U}}}$ similarly be the orthogonal projector onto the top-$M$ eigenspace of $\hat{M}_2$, and let $P_{\hat{\mathbf{U}}^{\perp}} = I - P_{\hat{\mathbf{U}}}$. By Weyl's theorem, we have $\lambda_{M+1}(\hat{M}_2) \leq \delta$ since $\text{rank}(M_2)$ is $M$. Our goal is to bound

$$\|\mu_m - P_{\hat{\mathbf{U}}} \mu_m\|_2^2 = \|P_{\hat{\mathbf{U}}^{\perp}} \mu_m\|_2^2 = \max_{y \in Range(\hat{\mathbf{U}}^{\perp}) \cap \mathbb{S}^{AZ-1}} (y^{\top} \mu_m)^2.$$

for any $m \in [M]$. Let $y$ be the vector maximizing the above. Since $w_m \mu_m \mu_m^{\top} \preceq M_2$ and since $y$ belongs to the eigenspace of ranks lower than $M$,

$$w_m y^{\top} \mu_m \mu_m^{\top} y \leq y^{\top} M_2 y = y^{\top} (M_2 - \hat{M}_2) y + y^{\top} \hat{M}_2 y \leq \|M_2 - \hat{M}_2\|_2 + \delta \leq 2\delta.$$

We have shown that $\|\mu_m - \hat{\mathbf{U}} \mu_m\|_2 \leq \sqrt{2\delta/w_m}$. Thus we need $\delta := O(\delta_{\text{sub}}^2)$ to bound the $l_2$ error by $\delta_{\text{sub}}/w_m^{1/2}$. Thus $N_0 = O\left(Z^2 A^4 \log(AZ/\eta)/\delta_{\text{sub}}^4\right)$ samples are sufficient for the estimation of the subspace $\mathbf{U}$. Note that we have not optimized for the polynomial factors which can be improved by more tightly bounding $\delta = \|M_2 - \hat{M}_2\|_2$ from $N_0$ samples.

## C.8 Deferred Details of Theorem 3.8

The proof of Theorem 3.8 follows by collecting the results from the preceding lemmas for subspace estimation (Lemma 3.7) and results from Section 3.2 and 3.3. Before we get into the proof, let us describe some details about finding a set of core action-value pairs, and how to recover the original model parameters $\{\mu_m\}_{m=1}^M$ from $\{\nu_m\}_{m=1}^M$.

### C.8.1 Detailed Procedures for Experimental Design

**Step 1. Subspace estimation**  We first find the set of core action-value pairs $\{(a_j, z_j)\}_{j=1}^n$ following the same procedure in Corollary 3.2. By Lemma 3.7, for every $\mu_m$, we have

$$\mu_m + \Delta_m \in \widehat{\mathbf{U}},$$

where $\|\Delta_m\|_\infty \le \delta_{\mathrm{sub}}/w_m^{1/2}$.

**Step 2. Pick core action-event pairs**  Let $\{\hat\beta_j\}_{j=1}^M$ be the orthonormal basis of $\widehat{\mathbf{U}}$. We can use this basis as input to Corollary 3.2 to get a set of (approximate) core action-event pairs. Specifically, let $\hat\Phi \in \mathbb{R}^{AZ \times M}$ be a feature matrix where each $j^{\text{th}}$ column $\hat\Phi_{:,j}$ is given as:

$$\hat\Phi_{:,j}(a, z) := \hat\beta_j(a, z), \qquad \forall a \in \mathcal{A}, z \in \mathcal{Z}. \tag{16}$$

After invoking Theorem B.1 with supplying $\hat\Phi$ as input, we use the support of $\rho$ as the set of core action-value pairs $\{(a_j, z_j)\}_{j=1}^n$. Let $\hat{G}(\rho)$ be defined as in equation (10). Note that with too small mixing weights $w_m$, we can instead use $\Delta_m = -\mu_m$ with $\|\Delta_m\|_\infty \le 1$.

**Step 3. Search constraints for moment-matching**  After finding a core action-event pairs $\{(a_j, z_j)\}_{j=1}^n$ as in Corollary 3.2, we estimate $\nu_m$ from higher-order tensors $\{\hat{T}_l\}_{l=1}^{\min(H, 2M-1)}$. When searching parameters for $\{(\hat{w}_m, \hat\nu_m)\}_{m=1}^M$, we can put constraints to ensure that $\hat{w}_m$ and $\hat\nu_m$ belong to a set of valid parameters for all $m \in [M]$:

$$\sum_{m=1}^M \hat{w}_m = 1, \quad w_{\min} \le \hat{w}_m, \quad 0 \preceq \hat\nu_m \preceq 1,$$

$$\left| \sum_{z \in \mathcal{Z}} (\hat{T}\hat\nu_m)(a, z) - 1 \right| \le -2Z\sqrt{M}\delta_{\mathrm{sub}}/\hat{w}_m^{1/2},$$

$$-2\sqrt{M}\delta_{\mathrm{sub}}/\hat{w}_m^{1/2} \preceq \hat{T}\hat\nu_m \preceq 1 + 2\sqrt{M}\delta_{\mathrm{sub}}/\hat{w}_m^{1/2}, \qquad \forall m \in [M], a \in \mathcal{A}, \tag{17}$$

where $\preceq$ is an element-wise inequality. That is, we want that $\hat{v}_m = \hat{T}\hat\nu_m$ is not too far from $\hat\mu_m$ after clipping and normalization. Without loss of generality, we assume that a rough estimate of $w_{\min}$ is known in advance (otherwise, we can repeat the same procedure with geometrically decreasing estimates of $w_{\min}$, e.g., $1/M, 1/2M, 1/4M, ..., 1/M^{O(\min(M,H))}$, and pick the best returned policy). A solution satisfying all constraints is guaranteed to exist since the true model $\{(w_m, \nu_m)\}_{m=1}^M$ also satisfies constraints.

**Step 4. Recovery of parameters**  Let $\hat\mu_m$ be computed by (11) using $\hat\nu_m$ and $\hat\Phi$, i.e., let $\hat{T} \in \mathbb{R}^{AZ \times n}$ be defined as

$$\hat{T}_{:,j} := \rho(a_j, z_j)\hat\Phi\hat{G}(\rho)^{-1}\hat\Phi_{(a_j, z_j),:} \qquad \forall j \in [n]. \tag{18}$$

We let $v_m = \hat{T}\nu_m, \hat{v}_m = \hat{T}\hat\nu_m$. We recover $\hat\mu_m$ from $\hat{v}_m = \hat{T}\hat\nu_m$ as

$$\tilde\mu_m := \mathrm{clip}(\hat{v}_m, 0, 1),$$

$$\hat\mu_m(a, z) := \frac{\tilde\mu_m(a, z)}{\sum_{z' \in \mathcal{Z}} \tilde\mu_m(a, z')}, \qquad \forall a \in \mathcal{A}, z \in \mathcal{Z}. \tag{19}$$

A simple algebra can show that the normalized estimates of $\hat\mu_m$ are close to $\mu_m$. Specifically, we show the following with Proposition B.2:

$$\|\mu_m - \hat{v}_m\|_\infty \le \|\Delta_m\|_\infty + (\|\Delta_m\|_\infty + \max_{j \in [n]} |\mu_m(a_j, z_j) - \hat\mu_m(a_j, z_j)|) \cdot \sqrt{2M}$$

$$\le \|\Delta_m\|_\infty + (\|\Delta_m\|_\infty + \|\nu_m - \hat\nu_m\|_\infty) \cdot \sqrt{2M}.$$

### C.8.2 Proof of Theorem 3.8, Case I: $H \geq 2M - 1$

We search the empirical parameters $\{(\hat{w}_m, \hat{\nu}_m)\}_{m=1}^M$) over the set (17). After finding $\hat{w}_m, \hat{\nu}_m$ that satisfy the moment matching condition (7), we recover $\hat{\mu}_m$ from $\hat{v}_m = \hat{T}\hat{\nu}_m$ after clipping and normalization (19). Then we first observe that for any $a \in \mathcal{A}$,

$$\|\mu_m(a, \cdot) - \tilde{\mu}_m(a, \cdot)\|_1 \leq \sum_{z \in \mathcal{Z}} |\mu_m(a, z) - \hat{v}_m(a, z)| \leq Z \max_{z \in \mathcal{Z}} |\mu_m(a, z) - \hat{v}_m(a, z)|,$$

where in the first inequality, we used the fact that clipping can only improve the $l_1$ error. Then, errors from the normalization can be bounded as

$$\begin{aligned}
\|\mu_m(a, \cdot) - \hat{\mu}_m(a, \cdot)\|_1 &\leq \|\mu_m(a, \cdot) - \tilde{\mu}_m(a, \cdot)\|_1 + \|\tilde{\mu}_m(a, \cdot) - \hat{\mu}_m(a, \cdot)\|_1 \\
&= \|\mu_m(a, \cdot) - \tilde{\mu}_m(a, \cdot)\|_1 + |\|\tilde{\mu}_m(a, \cdot)\|_1 - 1| \\
&\leq 2\|\mu_m(a, \cdot) - \tilde{\mu}_m(a, \cdot)\|_1 \\
&\leq 2Z \max_{z \in \mathcal{Z}} |\mu_m(a, z) - \hat{v}_m(a, z)| \\
&\leq 2Z \left( \|\Delta_m\|_\infty + (\|\Delta_m\|_\infty + \|\nu_m - \hat{\nu}_m\|_\infty)\sqrt{2M} \right).
\end{aligned}$$

The second line holds since

$$\begin{aligned}
\|\tilde{\mu}_m(a, \cdot) - \hat{\mu}_m(a, \cdot)\|_1 &= \sum_{z \in \mathcal{Z}} \left| \tilde{\mu}_m(a, z) - \frac{\tilde{\mu}_m(a, z)}{\sum_{z \in \mathcal{Z}} \tilde{\mu}_m(a, z)} \right| \\
&= \left| 1 - \frac{1}{\sum_{z \in \mathcal{Z}} \tilde{\mu}_m(a, z)} \right| \cdot \sum_{z \in \mathcal{Z}} |\tilde{\mu}_m(a, z)| \\
&= \left| \sum_{z \in \mathcal{Z}} \tilde{\mu}_m(a, z) - \frac{\sum_{z \in \mathcal{Z}} \tilde{\mu}_m(a, z)}{\sum_{z \in \mathcal{Z}} \tilde{\mu}_m(a, z)} \right| = |\|\tilde{\mu}_m(a, z)\|_1 - 1|.
\end{aligned}$$

where the third relation holds since $\sum_{z \in \mathcal{Z}} |\tilde{\mu}_m(a, z)| / |\sum_{z \in \mathcal{Z}} \tilde{\mu}_m(a, z)| = 1$ since $\tilde{\mu}_m(a, z) \geq 0$.

By the choice of $\delta_{\text{sub}}$, we have $\|\Delta_m\|_\infty \leq \delta_{\text{sub}}/w_m^{1/2} \leq \epsilon/(2ZMH^2 w_m^{1/2})$. Now we can call Proposition 3.4, and proceed as

$$\begin{aligned}
|V(\pi) - \hat{V}(\pi)| &\leq H^2 \cdot \inf_\Gamma \sum_{(m,m') \in [M]^2} \left( \Gamma(m, m') \cdot \max_{a \in \mathcal{A}} \|\mu_m(a, \cdot) - \hat{\mu}_{m'}(a, \cdot)\|_1 \right) \\
&\leq 2ZH^2 \cdot \inf_\Gamma \sum_{(m,m') \in [M]^2} \Gamma(m, m') \cdot \left( \sqrt{M/w_m} \cdot \epsilon/(2ZMH^2) + \sqrt{2M} \cdot \|\nu_m - \hat{\nu}_{m'}\|_\infty \right) \\
&\leq 2ZH^2 \cdot \left( \sum_{m \in [M]} w_m \cdot \sqrt{M/w_m} \cdot \epsilon/(2ZMH^2) + \sqrt{2M} \cdot W(\gamma, \hat{\gamma}) \right) \\
&\leq 2ZH^2 \cdot \left( \epsilon/(2H^2) + 2\sqrt{2M} \cdot W(\gamma, \hat{\gamma}) \right),
\end{aligned}$$

where in the last inequality, we used Cauchy-Schwartz inequality $\sum_{m=1}^M \sqrt{w_m} \leq \sqrt{M}$. Hence if we have $2ZH^2\sqrt{2M}W(\gamma, \hat{\gamma}) \leq \epsilon$, which is given by the choice of $\delta_{\text{sub}}$ and Lemma 3.3, we have $|V(\pi) - \hat{V}(\pi)| \leq O(\epsilon)$.

### C.8.3 Proof of Theorem 3.8, Case II: $H < 2M - 1$

If $H < 2M - 1$, we start by observing that for any $0 \preceq \hat{\nu}_m \preceq 1$, using Proposition B.2 and setting $\mu = 0$, $\epsilon_0 = 0$, $\eta = \nu_m$ and $\epsilon_1 = 1$, we have $-\sqrt{2M} \preceq \hat{T}\hat{\nu}_m \preceq \sqrt{2M}$.

To exploit the moment-closeness property, we define an auxiliary model $\{(w_m, v_m)\}_{m=1}^M$ where $v_m := \hat{T}\nu_m$. Similarly to $H \geq 2M - 1$ case, for any $a \in \mathcal{A}$, we have that

$$\|v_m(a, \cdot) - \mu_m(a, \cdot)\|_1 \leq 2Z\|\Delta_m\|_\infty(1 + \sqrt{2M}) \leq Z\sqrt{2M}\delta_{\text{sub}}/w_m^{1/2}. \tag{20}$$

We also have that

$$\|v_m(a, \cdot)\|_1 \leq \min\left(Z\sqrt{2M}\delta_{\mathrm{sub}}/w_m^{1/2}, Z\sqrt{2M}\right). \tag{21}$$

Let $\hat{v}_m = \hat{T}\hat{\nu}_m$ and $\hat{\mu}_m$ be defined as in (19). Recall our goal to bound

$$V(\pi) - \hat{V}(\pi) = \sum_{a_{1:H}, r_{1:H}} \left(\sum_{t=1}^{H} r_t\right) \left(\sum_{m=1}^{M} w_m \Pi_{t=1}^{H} \mu_m(a_t, r_t) - \sum_{m=1}^{M} \hat{w}_m \Pi_{t=1}^{H} \hat{\mu}_m(a_t, r_t)\right) \pi(a_{1:H}|r_{1:H-1}).$$

Define auxiliary value functions $V_{\mathrm{aux}}(\pi)$ and $\hat{V}_{\mathrm{aux}}(\pi)$ as

$$V_{\mathrm{aux}}(\pi) := \sum_{a_{1:H}, r_{1:H}} \left(\sum_{t=1}^{H} r_t\right) \sum_{m=1}^{M} w_m \Pi_{t=1}^{H} v_m(a_t, r_t) \pi(a_{1:H}|r_{1:H-1}),$$

$$\hat{V}_{\mathrm{aux}}(\pi) := \sum_{a_{1:H}, r_{1:H}} \left(\sum_{t=1}^{H} r_t\right) \sum_{m=1}^{M} \hat{w}_m \Pi_{t=1}^{H} \hat{v}_m(a_t, r_t) \pi(a_{1:H}|r_{1:H-1}),$$

Then $|V(\pi) - \hat{V}(\pi)| \leq |V(\pi) - V_{\mathrm{aux}}(\pi)| + |V_{\mathrm{aux}}(\pi) - \hat{V}_{\mathrm{aux}}(\pi)| + |\hat{V}_{\mathrm{aux}}(\pi) - \hat{V}(\pi)|$. We bound each term separately.

**Term I.** $|V(\pi) - V_{\mathrm{aux}}(\pi)|$ : This is less than

$$|V(\pi) - V_{\mathrm{aux}}(\pi)| = \left| \sum_{a_{1:H}, r_{1:H}} \left(\sum_{t=1}^{H} r_t\right) \sum_{m=1}^{M} w_m \left(\Pi_{t=1}^{H} \mu_m(a_t, r_t) - \Pi_{t=1}^{H} v_m(a_t, r_t)\right) \pi(a_{1:H}|r_{1:H-1}) \right|$$

$$\leq H \cdot \sum_{a_{1:H}, r_{1:H}} \left| \sum_{m=1}^{M} w_m \left(\Pi_{t=1}^{H} \mu_m(a_t, r_t) - \Pi_{t=1}^{H} v_m(a_t, r_t)\right) \pi(a_{1:H}|r_{1:H-1}) \right|$$

$$\leq H \cdot \sum_{a_{1:H-1}, r_{1:H-1}} \sum_{m=1}^{M} w_m \left| \Pi_{t=1}^{H-1} \mu_m(a_t, r_t) - \Pi_{t=1}^{H-1} v_m(a_t, r_t) \right| \sum_{a_H, r_H} |v_m(a_H, r_H)| \pi(a_{1:H}|r_{1:H-1})$$

$$+ H \cdot \sum_{a_{1:H-1}, r_{1:H-1}} \sum_{m=1}^{M} w_m \Pi_{t=1}^{H-1} \mu_m(a_t, r_t) \sum_{a_H, r_H} |\mu_m(a_H, r_H) - v_m(a_H, r_H)| \pi(a_{1:H}|r_{1:H-1})$$

$$\leq H \sum_{m=1}^{M} w_m \max_{a \in \mathcal{A}} \|v_m(a, \cdot)\|_1 \cdot \sum_{a_{1:H-1}, r_{1:H-1}} \left| \Pi_{t=1}^{H-1} \mu_m(a_t, r_t) - \Pi_{t=1}^{H-1} v_m(a_t, r_t) \right| \pi(a_{1:H-1}|r_{1:H-2})$$

$$+ H \sum_{m=1}^{M} w_m \max_{a \in \mathcal{A}} \|\mu_m(a, \cdot) - v_m(a, \cdot)\|_1 \cdot \sum_{a_{1:H-1}, r_{1:H-1}} \Pi_{t=1}^{H-1} \mu_m(a_t, r_t) \pi(a_{1:H}|r_{1:H-1}).$$

For the second term, we have

$$\left| \sum_{a_{1:H-1}, r_{1:H-1}} \Pi_{t=1}^{H-1} \mu_m(a_t, r_t) \pi(a_{1:H}|r_{1:H-1}) \right| = \left| \sum_{a_{1:H-1}, r_{1:H-1}} \mathbb{P}_m(a_{1:H-1}, r_{1:H-1}) \right| = 1$$

for all $m \in [M]$. For the first term, we can recursively apply the same inequality until the time step reaches to $t = 1$. Applying this recursively,

$$|V(\pi) - V_{\mathrm{aux}}(\pi)| \leq H^2 \cdot \sum_{m=1}^{M} w_m \left(\max_{a \in \mathcal{A}} \|v_m(a, \cdot)\|_1\right)^{H-1} \cdot \max_{a \in \mathcal{A}} \|\mu_m(a, \cdot) - v_m(a, \cdot)\|_1.$$

To bound the above, we first consider the case when $w_m \geq \frac{\epsilon}{H^2 M (Z\sqrt{2M})^H}$. Note that if $w_{\min} \geq \frac{\epsilon}{H^2 M (Z\sqrt{2M})^H}$, then this is always the case. In this case, for every $a \in \mathcal{A}$, we have (recall (20))

$$\max_{a \in \mathcal{A}} \|\mu_m(a, \cdot) - v_m(a, \cdot)\|_1 \leq 2\sqrt{M} Z\delta_{\mathrm{sub}}/w_m^{1/2} \leq \frac{\epsilon}{H^2\sqrt{Mw_m}},$$

$$\max_{a \in \mathcal{A}} \|v_m(a, \cdot)\|_1 \le 1 + \max_{a \in \mathcal{A}} \|v_m(a, \cdot) - \mu_m(a, \cdot)\|_1$$
$$\le 1 + 2\sqrt{M} Z \delta_{\text{sub}}/w_m^{1/2} \le 1 + 1/H,$$

where we use our choice of $\delta_{\text{sub}}$ in (5). By the second condition,

$$(\max_{a \in \mathcal{A}} \|v_m(a, \cdot)\|_1)^{H-1} \le (1 + 1/H)^{H-1} \le e.$$

The first condition can be combined with Proposition B.2 similarly to the $H \ge 2M - 1$ case to get

$$H^2 \sum_{m : w_m \ge \frac{\epsilon}{H^2 M (Z\sqrt{2M})^H}} w_m \left( \max_{a \in \mathcal{A}} \|v_m(a, \cdot)\|_1 \right)^{H-1} \max_{a \in \mathcal{A}} \|\mu_m(a, \cdot) - v_m(a, \cdot)\|_1$$

$$\le e \sum_{m=1}^{M} \sqrt{w_m/M} \epsilon \le O(\epsilon).$$

On the other hand, if $w_m < \frac{\epsilon}{H^2 M (Z\sqrt{2M})^H}$, then we can directly bound as

$$H^2 \sum_{m : w_m < \frac{\epsilon}{H^2 M (Z\sqrt{2M})^H}} w_m \left( \max_{a \in \mathcal{A}} \|v_m(a, \cdot)\|_1 \right)^{H-1} \max_{a \in \mathcal{A}} \|\mu_m(a, \cdot) - v_m(a, \cdot)\|_1$$

$$\le \frac{\epsilon}{M} \sum_{m=1}^{M} \frac{1}{(Z\sqrt{2M})^H} (Z\sqrt{2M})^H \le O(\epsilon).$$

Thus, we have $|V(\pi) - V_{\text{aux}}(\pi)| \le O(\epsilon)$.

**Term II.** $|V_{\text{aux}}(\pi) - \hat{V}_{\text{aux}}(\pi)|$ : We use the moment-closeness properties between $v_m$ and $\hat{v}_m$ given similarly to Lemma 3.6.

**Lemma C.4** *For any given degree $l \ge 1$, if $\|\sum_{m=1}^{M} w_m \nu_m^{\otimes l} - \sum_{m=1}^{M} \hat{w}_m \hat{\nu}_m^{\otimes l}\|_\infty \le \delta$, then $v_m$ and $\hat{v}_m$ satisfy*

$$\left\| \sum_{m=1}^{M} w_m v_m^{\otimes l} - \sum_{m=1}^{M} \hat{w}_m \hat{v}_m^{\otimes l} \right\|_\infty \le (2M)^{l/2} \cdot \delta.$$

*Proof.* Note that $\|\sum_{m=1}^{M} w_m (\hat{T}\nu_m)^{\otimes l} - \sum_{m=1}^{M} \hat{w}_m (\hat{T}\hat{\nu}_m)^{\otimes l}\|_\infty \le (2M)^{l/2}\delta$, following the same argument in Appendix C.6: the conclusion of Proposition B.2 also implies that the $l_1$ norm of every row in $\hat{T}$ is less than $\sqrt{2M}$, *i.e.,*

$$\|\hat{T}_{(a,z),:}\|_1 \le \sqrt{2M}, \qquad \forall (a, z) \in \mathcal{A} \times \mathcal{Z}, \tag{22}$$

Lemma follows since $v_m$ is a vector consisting of partial coordinates of $\hat{T}\nu_m$. $\square$

Now we proceed as

$$|V_{\text{aux}}(\pi) - \hat{V}_{\text{aux}}(\pi)| = \left| \sum_{a_{1:H}, r_{1:H}} \left( \sum_{t=1}^{H} r_t \right) \left( \sum_{m=1}^{M} w_m \Pi_{t=1}^{H} v_m(a_t, r_t) - \sum_{m=1}^{M} \hat{w}_m \Pi_{t=1}^{H} \hat{v}_m(a_t, r_t) \right) \pi(a_{1:H}|r_{1:H-1}) \right|$$

$$\le H \cdot \sum_{a_{1:H}, r_{1:H}} \left| \sum_{m=1}^{M} w_m \Pi_{t=1}^{H} v_m(a_t, r_t) - \sum_{m=1}^{M} \hat{w}_m \Pi_{t=1}^{H} \hat{v}_m(a_t, r_t) \right| \pi(a_{1:H}|r_{1:H-1})$$

$$\le H Z^H (2M)^{H/2} \delta_{\text{tsr}}.$$

Choice of $\delta_{\text{tsr}} = (\epsilon/H)/(\sqrt{2M}Z)^H$ for $H < 2M - 1$ gives $|V_{\text{aux}}(\pi) - \hat{V}_{\text{aux}}(\pi)| = O(\epsilon)$.

**Term III.** $|\hat{V}_{\mathrm{aux}}(\pi) - \hat{V}(\pi)|$   : This case is almost similar to the case $|V(\pi) - V_{\mathrm{aux}}(\pi)|$.

$$|\hat{V}(\pi) - \hat{V}_{\mathrm{aux}}(\pi)| = \left| \sum_{a_{1:H}, r_{1:H}} \left( \sum_{t=1}^{H} r_t \right) \sum_{m=1}^{M} \hat{v}_m \left( \Pi_{t=1}^{H} \hat{\mu}_m(a_t, r_t) - \Pi_{t=1}^{H} \hat{\mu}_m(a_t, r_t) \right) \pi(a_{1:H}|r_{1:H-1}) \right|$$

$$\leq H \cdot \sum_{a_{1:H}, r_{1:H}} \left| \sum_{m=1}^{M} \hat{w}_m \left( \Pi_{t=1}^{H} \hat{v}_m(a_t, r_t) - \Pi_{t=1}^{H} \hat{\mu}_m(a_t, r_t) \right) \pi(a_{1:H}|r_{1:H-1}) \right|$$

$$\leq H \cdot \sum_{a_{1:H-1}, r_{1:H-1}} \sum_{m=1}^{M} \hat{w}_m \left| \Pi_{t=1}^{H-1} \hat{v}_m(a_t, r_t) - \Pi_{t=1}^{H-1} \hat{\mu}_m(a_t, r_t) \right| \sum_{a_H, r_H} |\hat{v}_m(a_H, r_H)| \pi(a_{1:H}|r_{1:H-1})$$

$$+ H \cdot \sum_{a_{1:H-1}, r_{1:H-1}} \sum_{m=1}^{M} \hat{w}_m \Pi_{t=1}^{H-1} \hat{\mu}_m(a_t, r_t) \sum_{a_H, r_H} |\hat{\mu}_m(a_H, r_H) - \hat{v}_m(a_H, r_H)| \pi(a_{1:H}|r_{1:H-1})$$

$$\leq H \cdot \sum_{a_{1:H-1}, r_{1:H-1}} \sum_{m=1}^{M} \hat{w}_m \max_{a \in \mathcal{A}} \|\hat{v}_m(a, \cdot)\|_1 \left| \Pi_{t=1}^{H-1} \hat{\mu}_m(a_t, r_t) - \Pi_{t=1}^{H-1} \hat{v}_m(a_t, r_t) \right| \pi(a_{1:H-1}|r_{1:H-2})$$

$$+ H \sum_{m=1}^{M} \hat{w}_m \max_{a \in \mathcal{A}} \|\hat{\mu}_m(a, \cdot) - \hat{v}_m(a, \cdot)\|_1 \sum_{a_{1:H-1}, r_{1:H-1}} \Pi_{t=1}^{H-1} \hat{\mu}_m(a_t, r_t) \pi(a_{1:H}|r_{1:H-1})$$

$$\leq H^2 \cdot \left( \sum_{m=1}^{M} \hat{w}_m \left( \max_{a \in \mathcal{A}} \|\hat{v}_m(a, \cdot)\|_1 \right)^{H-1} \cdot \max_{a \in \mathcal{A}} \|\hat{\mu}_m(a, \cdot) - \hat{v}_m(a, \cdot)\|_1 \right).$$

For each $m \in [M]$, if $\hat{w}_m \geq \frac{\epsilon}{H^2 M (Z\sqrt{2M})^H}$, then

$$\|\hat{v}_m(a, \cdot)\|_1 \leq \left| \sum_{z: \hat{v}_m(a,z) < 0} \hat{v}_m(a, z) \right| + \left| \sum_{z: \hat{v}_m(a,z) > 0} \hat{v}_m(a, z) \right|$$

$$\leq 1 + 2Z\sqrt{M} \delta_{\mathrm{sub}}/\hat{w}_m^{1/2} \leq 1 + 1/H,$$

where we used the choice of $\delta_{\mathrm{sub}}$ in (5). We also need to show that $\|\hat{v}_m(a, \cdot) - \hat{\mu}_m(a, \cdot)\|_1$ is bounded. Let $\tilde{\mu}_m$ be the intermediate step after clipping $\hat{v}_m$ before normalization as in 19. Due to the third condition of (17), clipped amount can be at most

$$\|\hat{v}_m(a, \cdot) - \tilde{\mu}_m(a, \cdot)\|_1 \leq 2Z\sqrt{M} \delta_{\mathrm{sub}}/\hat{w}_m^{1/2}.$$

With the second condition of (17), we have

$$\|\tilde{\mu}_m(a, \cdot)\|_1 = \sum_{z \in \mathcal{Z}} \tilde{\mu}_m(a, z) \leq \left| \sum_{z \in \mathcal{Z}} \tilde{\mu}_m(a, z) - \hat{\mu}_m(a, z) \right| + \left| \sum_{z \in \mathcal{Z}} \hat{\mu}_m(a, z) \right|$$

$$\leq \|\hat{v}_m(a, \cdot) - \tilde{\mu}_m(a, \cdot)\|_1 + \|\hat{\mu}_m(a, \cdot)\|_1 \leq 1 + 3Z\sqrt{M} \delta_{\mathrm{sub}}/\hat{w}_m^{1/2}.$$

Similarly,

$$\|\tilde{v}_m(a, \cdot)\|_1 = \sum_{z \in \mathcal{R}} \tilde{v}_m(a, z) \geq \left| \sum_{z \in \mathcal{R}} v_m(a, z) \right| - \left| \sum_{z \in \mathcal{R}} \tilde{v}_m(a, z) - v_m(a, z) \right|$$

$$\geq |\hat{\phi}_m(a, \mathcal{R})| - \|\hat{v}_m(a, \cdot) - \tilde{v}_m(a, \cdot)\|_1 \geq 1 - 3Z\sqrt{M} \delta_{\mathrm{sub}}/\hat{w}_m^{1/2}.$$

Therefore, we can show that

$$\|\hat{\mu}_m(a, \cdot) - \hat{v}_m(a, \cdot)\|_1 \leq \|\hat{\mu}_m(a, \cdot) - \tilde{v}_m(a, \cdot)\|_1 + \|\tilde{v}_m(a, \cdot) - \hat{v}_m(a, \cdot)\|_1$$

$$\leq \frac{|\|\tilde{v}_m(a, \cdot)\|_1 - 1|}{\|\tilde{v}_m(a, \cdot)\|_1} + 2Z\sqrt{M} \delta_a/\hat{w}_m^{1/2}$$

$$\leq 8Z\sqrt{M} \delta_{\mathrm{sub}}/\hat{w}_m^{1/2},$$

where we used $1 - 3Z\delta_{\mathrm{sub}}/\hat{w}_m^{1/2} \geq 1/2$ due to the first constraint and the choice of $\delta_{\mathrm{sub}}$.

If $\hat{w}_m < \frac{\epsilon}{H^2 M (Z\sqrt{2M})^H}$, then we can use the fact that all $l_1$-norm of rows of $\hat{T}$ are less than $\sqrt{2M}$ (equation (22)), and thus $\|\hat{v}_m(a,\cdot)\|_1 \le \sqrt{2M}$ for all $a \in \mathcal{A}$. We also have that $\|\hat{\mu}_m(a,\cdot) - \hat{v}_m(a,\cdot)\|_1 \le 1 + \sqrt{2M}$. Now we plug all things together, and proceed as

$$
\begin{aligned}
|\hat{V}(\pi) - \hat{V}_{\mathrm{aux}}(\pi)| &\le H^2 \cdot \left( \sum_{m=1}^{M} \hat{w}_m \left( \max_{a \in \mathcal{A}} \|\hat{v}_m(a,\cdot)\|_1 \right)^{H-1} \cdot \max_{a \in \mathcal{A}} \|\hat{\mu}_m(a,\cdot) - \hat{v}_m(a,\cdot)\|_1 \right) \\
&\le e H^2 \cdot \left( \sum_{m:\hat{w}_m \ge \frac{\epsilon}{H^2 M (Z\sqrt{2M})^H}} \hat{w}_m \cdot \max_{a \in \mathcal{A}} \|\hat{\mu}_m(a,\cdot) - \hat{v}_m(a,\cdot)\|_1 \right) \\
&\quad + H^2 \cdot \left( \sum_{m:\hat{w}_m < \frac{\epsilon}{H^2 M (Z\sqrt{2M})^H}} \hat{w}_m \cdot (Z\sqrt{2M})^H \right) \\
&\le 4 e H^2 \sum_{m=1}^{M} \epsilon \sqrt{\hat{w}_m/M}/H^2 + \epsilon \le O(\epsilon).
\end{aligned}
$$

Collecting all three terms, we have $|V(\pi) - \hat{V}(\pi)| \le O(\epsilon)$. This concludes the proof of Theorem 3.8.

## Appendix D  LMAB with Gaussian Rewards

So far we have focused on rewards with finite support $Z = O(1)$. In this section we consider and LMAB setting with Gaussian rewards and generalize Algorithm 1. Indeed, some steps in the algorithm cannot be straightforwardly extended if $Z = \infty$. In this subsection, we consider a standard Gaussian reward distribution–a special case of continuous rewards–and generalize Algorithm 1 to this setting. We make the following assumption.

**Assumption D.1 (Gaussian Rewards)** *The reward distribution conditioning on an action $a \in \mathcal{A}$ in a context $m \in [M]$ is $\mathcal{N}(\mu_m(a), 1)$ for some $|\mu_m(a)| \le 1$.*

Even though the rewards have infinite support, we show that the same conclusion holds as in finite-support case, *i.e.,* the sample-complexity is upper bounded by $O((MH/\epsilon)^{O(\min(H,M))} + \mathrm{poly}(A, H, M))$. Algorithms for the Gaussian case differ significantly in identifiable $H \ge 2M - 1$ and unidentifiable regimes $H < 2M - 1$. When $H \ge 2M - 1$, there are only minor changes in the algorithm design for defining core actions and how tensors are constructed. We handle this case in Appendix D.1. A more interesting case is the parameter unidentifiable regime, where we follow an alternative approach and discretize the support of rewards by $O(\epsilon/H^2)$-level. This approach is described in Appendix D.2.

We can reach similar conclusions for the Gaussian rewards:

**Theorem D.2** *Consider any LMAB with $M$ contexts under Gaussian reward Assumption D.1. There exists an algorithm such that with probability at least $1 - \eta$, it returns an $\epsilon$-optimal policy using a number of episodes at most*

$$
\mathrm{poly}(H, M, A, 1/\epsilon, \log(A/\eta)) + \mathrm{poly}(\log(M/\eta), H, M)^{2M-1} \cdot \epsilon^{-(4M-2)}, \quad \textit{if } H \ge 2M - 1,
$$
$$
\mathrm{poly}(H, w_{\min}^{-1}, A, 1/\epsilon, \log(A/\eta)) + \mathrm{poly}(\log(MH/(\eta\epsilon)), H, M)^{H} \cdot \epsilon^{-(2H+2)}, \quad \textit{otherwise.}
$$

Note that the dependency on $\epsilon$ is at most $\epsilon^{-(4M-2)}$ and smaller when $H < 2M - 2$. In the parameter identifiable regime, similarly to discrete reward cases, near-optimality of returned policy comes from the closeness of latent model parameters in the Wasserstein metric. In the parameter unidentifiable regime, we first discretize the support of rewards in $O(\epsilon)$ level. Then we can apply the same procedures for handling discrete rewards as in Section 3.3. We provide the full details in Appendix D.

### D.1 Algorithm for Identifiable Regime $H \geq 2M - 1$

Let $\hat{M}_2 \in \mathbb{R}^{A \times A}$ be the empirical second-order moments as

$$\hat{M}_2 = \frac{1}{2N_0} \sum_{k=1}^{N_0} r_1^k r_2^k \cdot \boldsymbol{e}_{a_1^k} \boldsymbol{e}_{a_2^k}^\top.$$

Then let $\widehat{\mathbf{U}}$ be the subspace spanned by top-$M$ eigenvectors of $\hat{M}_2$. A similar conclusion to Lemma 3.7 holds:

**Lemma D.3** *Let $\widehat{\mathbf{U}}$ be a subspace spanned by top-$M$ eigenvectors of $\hat{M}_2$. After we estimate $\hat{M}_2$ using $N_0 = O(A^4 \log(A/\eta)/\delta_{\text{sub}}^4)$ episodes, with probability at least $1 - \eta$, for all $m \in [M]$, there exists $\Delta_m : \|\Delta_m\|_\infty \leq \delta_{\text{sub}}/w_m^{1/2}$ such that $\mu_m + \Delta_m \in \widehat{\mathbf{U}}$.*

Proof of Lemma D.3 is identical to the proof of Lemma 3.7. Let $\delta_{\text{sub}} = \epsilon/(2MH^2)$.

Similarly to finite-support reward distributions, let $\{\hat{\beta}_j\}_{j=1}^M$ be the orthonormal basis of $\widehat{\mathbf{U}}$ and construct $\hat{\Phi} \in \mathbb{R}^{A \times M}$ such that the $j^{\text{th}}$ column of $\hat{\Phi}$ is $\hat{\beta}_j$, *i.e.*, $\hat{\Phi}_{:,j} = \hat{\beta}_j$. We invoke Theorem B.1 to get a set of core actions $\{a_j\}_{j=1}^n$. The main difference to the finite-support case is that we do not need to specify a corresponding event of rewards. Instead, we measure the correlation in terms of actual reward values. Specifically, for every multi-index $(i_1, i_2, ..., i_l) \in [n]^l$, using $N_1$ episodes where

$$N_1 = O\left(\log(ln^l/\eta)\right)^l /\delta_{\text{tsr}}^2.$$

We play $a_t^k = a_{i_t}$ for $t = 1, ..., l$ and $k \in [N_b]$, and estimate higher-order moments:

$$\hat{T}_l(i_1, ..., i_l) = \frac{1}{N_1} \sum_{k=1}^{N_1} \Pi_{t=1}^d r_t^k.$$

We can easily verify that

$$T_l = \mathbb{E}[\hat{T}_l] = \sum_{m=1}^M w_m \nu_m^{\otimes l}.$$

We can apply the concentration of higher-order polynomials of sub-Gaussian random variables element-wise, given from the following lemma on hypercontractivity inequality:

**Lemma D.4 (Hypercontractivity Inequality (Theorem 1.9 in [34]))** *Consider a degree-$l$ polynomial $f$ defined over a set of $N$ independent samples of zero-mean unit-variance Gaussians, such that $f(X_{1:N}) := f(X_1, ..., X_N)$. Then,*

$$\mathbb{P}\left(|f(X_{1:N}) - \mathbb{E}[f(X_{1:N})]| \geq \lambda\right) \leq e^2 \exp\left(-\left(\frac{\lambda^2}{C \cdot var(f(X_{1:N}))}\right)^{1/l}\right),$$

*for some absolute constant $C > 0$.*

To show the concentration of $\hat{T}_l(i_1, ..., i_l)$ around $T_l(i_1, ..., i_l)$, we can apply Lemma D.4 with plugging $\lambda = O(\delta_{\text{tsr}})$ and $var(f) \leq 2^l \cdot var(X^l)/N_1$, where $X \sim \mathcal{N}(0, 1)$. Here, $f$ can be viewed as a degree-$l$ polynomial of $X_t^k := r_t^k - \mu_{m^k}(a_t^k)$ for $t = 1, \ldots, l$ and $k = 1, ..., N_1$, where $m^k$ is a latent context for the episode $k$.

Since $var(X^l) \leq O(l^l)$, we need $N_1 = O(l^l \log^l (ln^l/\eta)/\delta_{\text{sub}}^2)$ to make the exponent less than $\eta/(ln^l)$. Take union bound over all elements in $\hat{T}_l$ ensures that $\|T_l - \hat{T}_l\|_\infty \leq \delta_{\text{tsr}}$ with probability at least $1 - \eta$.

Now we find a set of parameters $\{(\hat{w}_m, \hat{\nu}_m)\}_{m=1}^M$ with the only constraint:

$$\hat{w}_m \in \mathbb{R}_+, \ \sum_{m=1}^M \hat{w}_m = 1.$$

We aim to find

$$\left\| \sum_{m=1}^{M} \hat{w}_m \hat{\nu}_m^{\otimes\, l} - \hat{T}_l \right\|_\infty \le \delta_{\text{tsr}}, \qquad \forall l \in [2M-1].$$

Now we can construct an empirical model by recovering $\hat{\mu}_m = \hat{T}\hat{\nu}_m$ where $\hat{T} \in \mathbb{R}^{A \times M}$ is defined similarly to (18) as

$$\hat{T}_{:,j} := \rho(a_j)\hat{\Phi}\hat{G}(\rho)^{-1}\hat{\Phi}_{a_j,:}, \qquad \forall j \in [n],$$

where $\rho$ is a distribution over rows of $\hat{\Phi}$ found by Theorem B.1 and $\hat{G}(\rho)$ is defined as in (10). We do not need extra clipping and normalization steps here since any $\hat{\mu}_m$ is a valid model parameter. Now with $\{(\hat{w}_m, \hat{\mu}_m)\}$, we call the planning oracle 2.3 and obtain an $\epsilon$-optimal policy.

## D.2 Algorithm with Short Time-Horizon $H < 2M - 1$

We first discretize possible reward values: let $\mathcal{Z} = \{z_1, z_2, ..., z_Z\}$ where $z_i = -4\sqrt{\log(H/\epsilon)} + (i-1) \cdot \epsilon/H^2$ and $Z = \lfloor 8H^2\sqrt{\log(H/\epsilon)}/\epsilon \rfloor$. We define an auxiliary reward (pseudo) p.d.f $p_m(a, \cdot)$ for each $a$ and $m$ as the following: for all $s \in [Z-1]$,

$$p_m(a, r) = \frac{H^2}{\epsilon} \int_{z_s}^{z_{s+1}} \frac{1}{\sqrt{2\pi}} \exp\left(-\frac{(x - \mu_m(a))^2}{2}\right) dx, \qquad \forall r \in [z_s, z_{s+1}), \quad (23)$$

and $p_m(a, r) = 0$ for all $r \in (-\infty, z_1) \cup [z_Z, \infty)$. Define $\tilde{V}(\cdot)$ be the policy evaluation function in a (pseudo) LMAB model $\tilde{\mathcal{B}} = (\mathcal{A}, \{w_m\}_{m=1}^M, \{p_m\}_{m=1}^M)$:

$$\tilde{V}(\pi) := \sum_{m=1}^{M} w_m \cdot \sum_{a_{1:H}} \int_{r_{1:H}} \left(\sum_{t=1}^{H} r_t\right) \Pi_{t=1}^{H} p_m(a_t, r_t)\pi(a_{1:H}|r_{1:H-1})d(r_{1:H}).$$

We first show that $p_m(a, \cdot)$ is good approximation of true reward distributions:

**Lemma D.5** *Let $\mathcal{B}$ and $\tilde{\mathcal{B}}$ defined as above. Then for any history-dependent policy $\pi$, $|V(\pi) - \tilde{V}(\pi)| \le 10\epsilon$.*

Given Lemma D.5, we will discretize reward values are so that we can leverage the result of Section 3.3 for short time-horizon. Specifically, let $\overline{\mathcal{B}}$ be a model $\{(w_m, q_m)\}_{m=1}^M$ with discrete reward distributions taking values in $\mathcal{Z} \cup \{0\}$, and let $q_m(a, z) = \frac{\epsilon}{H^2} \cdot p_m(a, z)$ for $z \in \mathcal{Z}$ and $q_m(a, 0) = \mathbb{P}_m(r \notin [z_1, z_Z]|a)$. As if the underlying model is $\overline{\mathcal{B}}$, we run Algorithm 1 with manually modifying the observed rewards $r_t \to \overline{r}_t$:

$$\overline{r}_t = 0, \qquad\qquad\qquad\qquad \text{if } r_t < z_1 \text{ or } r_t \ge z_Z,$$
$$\overline{r}_t = z_s, \qquad\qquad \text{for some } s \in [Z-1], \text{ s.t. } r_t \in [z_s, z_{s+1}).$$

From actions $(a_1, ..., a_H)$ and reward observations $(\overline{r}_1, ..., \overline{r}_H)$, we can now apply Algorithm 1 for the parameter unidentifiable case $H < 2M - 1$.

## D.3 Proof of Theorem D.2

Now we are ready to prove the Theorem D.2 for Gaussian rewards.

### D.3.1 Identifiable Regime $H \ge 2M - 1$:

By Lemma 3.3, we know that $W(\gamma, \hat{\gamma}) \le O\left(M^3 n \delta_{\text{tsr}}^{-1/(2M-1)}\right)$ where $\nu = \sum_{m=1}^{M} w_m \delta_{\nu_m}$ and $\hat{\nu} = \sum_{m=1}^{M} \hat{w}_m \delta_{\hat{\nu}_m}$. With experimental design, by Corollary 3.2, we can ensure that

$$\max_{a \in \mathcal{A}} |\mu_m(a) - \hat{\mu}_{m'}(a)| \le \sqrt{2M}\|\nu_m - \hat{\nu}_{m'}\|_\infty.$$

Observe that for any $a \in \mathcal{A}$, total variation distance between standard Gaussians is bounded by the distance between centers of Gaussians, *i.e.*, $d_{TV}(\mathcal{N}(\mu_m(a),1), \mathcal{N}(\hat{\mu}_{m'}(a),1)) \leq |\mu_m(a) - \hat{\mu}_{m'}(a)|$. Using Proposition 3.4, we can show that

$$
\begin{aligned}
|V(\pi) - \hat{V}(\pi)| &\leq 2H^2 \cdot \inf_{\Gamma} \sum_{(m,m')} \Gamma(m,m') \max_{a \in \mathcal{A}} d_{TV}(\mathcal{N}(\mu_m(a),1), \mathcal{N}(\hat{\mu}_{m'}(a),1)) \\
&\leq 2H^2 \cdot \inf_{\Gamma} \sum_{(m,m')} \Gamma(m,m') \max_{a \in \mathcal{A}} |\mu_m(a) - \hat{\mu}_{m'}(a)| \\
&\leq 2\sqrt{2M}H^2 \cdot \inf_{\Gamma} \sum_{(m,m')} \Gamma(m,m') \|\nu_m - \hat{\nu}_{m'}\|_\infty \\
&\leq 2\sqrt{2M}H^2 W(\gamma, \hat{\gamma}).
\end{aligned}
$$

Plugging the choice of $\delta_{\mathrm{tsr}} = O(\epsilon/(H^2 M^{3.5} n))^{2M-1}$, this is less than $\epsilon$.

### D.3.2 Unidentifiable Regime $H < 2M - 1$:

Let us first compare the expected rewards from $\tilde{\mathcal{B}}$ and $\overline{\mathcal{B}}$ with any fixed policy $\pi$.

$$
\tilde{V}(\pi) = \sum_{m=1}^M w_m \cdot \sum_{a_{1:H}} \int_{r_{1:H}} \left( \sum_{t=1}^H r_t \right) \Pi_{t=1}^H p_m(a_t, r_t) \pi(a_{1:H}|r_{1:H-1}) d(r_{1:H}),
$$

$$
\overline{V}(\pi) = \sum_{m=1}^M w_m \cdot \sum_{a_{1:H}} \sum_{\overline{r}_{1:H}} \left( \sum_{t=1}^H \overline{r}_t \right) \Pi_{t=1}^H q_m(a_t, \overline{r}_t) \pi(a_{1:H}|\overline{r}_{1:H-1}).
$$

With slight abuse in notation, let $\overline{r}_t$ be a quantized value of $r_t$. Then we can show that

$$
\begin{aligned}
\left| \tilde{V}(\pi) - \overline{V}(\pi) \right| &\leq \sum_{t=1}^H \int_{r_{1:H}:r_t \in [z_1,z_Z], \forall t \in [H]} \left| \sum_{m=1}^M w_m \sum_{a_{1:H}} (r_t - \overline{r}_t) \Pi_{t=1}^H p_m(a_t,r_t) \pi(a_{1:H}|r_{1:H-1}) d(r_{1:H}) \right| \\
&\quad + \sum_{t=1}^H \int_{r_{1:H}:r_t \notin [z_1,z_Z], \exists t \in [H]} \left| \sum_{m=1}^M w_m \sum_{a_{1:H}} \overline{r}_t \Pi_{t=1}^H q_m(a_t,r_t) \pi(a_{1:H}|r_{1:H-1}) d(r_{1:H}) \right| \\
&\leq \sum_{t=1}^H \int_{r_{1:H}:r_t \in [z_1,z_Z], \forall t \in [H]} \sum_{m=1}^M w_m \sum_{a_{1:H}} (\epsilon/H^2) \Pi_{t=1}^H p_m(a_t,r_t) \pi(a_{1:H}|r_{1:H-1}) d(r_{1:H}) \\
&\quad + |Hz_Z| \cdot \mathbb{P}(\exists t \in [H], \ s.t. \ r_t \notin [z_1,z_Z]) \\
&\leq \epsilon/H + |H^2 z_Z| \cdot \mathbb{P}_{X \sim \mathcal{N}(0,1)}(X \geq z_Z - 1),
\end{aligned}
$$

where in the first inequality, we used $p_m(a,r) = 0$ for $r \notin [z_1, z_Z]$. Note that $\mathbb{P}_{X \sim \mathcal{N}(0,1)}(|X| \geq z_Z - 1) \leq (H/\epsilon)^4$ with $z_Z = 4\sqrt{\log(H/\epsilon)}$.

Note that a system with manually discretized rewards can be described by the model $\overline{\mathcal{B}}$. Assuming the returned policy $\hat{\pi}$ from Algorithm 1 is $O(\epsilon)$-optimal for $\overline{\mathcal{B}}$, by triangle inequality for the policy evaluation for any policy $\pi$,

$$
|\overline{V}(\pi) - V(\pi)| \leq |\overline{V}(\pi) - \tilde{V}(\pi)| + |\tilde{V}(\pi) - V(\pi)|,
$$

we conclude that $\hat{\pi}$ is $O(\epsilon)$-optimal for $\mathcal{B}$ with Gaussian rewards.

### D.4 Proof of Lemma D.5

We start by unfolding the expression for policy value differences.

$$
|f(\pi) - \tilde{f}(\pi)| \leq \sum_{m=1}^M w_m \cdot \sum_{a_{1:H}} \int_{r_{1:H}} \left( \sum_{t=1}^H |r_t| \right) \left| \Pi_{t=1}^H p_m(a_t,r_t) - \Pi_{t=1}^H g_m(a_t,r_t) \right| \pi(a_{1:H}|r_{1:H-1}) d(r_{1:H}),
$$

where $g_m(a_t, r_t) := \frac{1}{\sqrt{2\pi}} \exp\left(-(r_t - \mu_m(a_t))^2/2\right)$. We first rule out reward values greater than $z_Z = 4\sqrt{\log(H/\epsilon)}$. Define a set of bad reward sequences $\mathcal{E}_b = \{r_{1:H} | \exists t \in [H], s.t., |r_t| > z_Z\}$. Then for any $t_0 \in [H]$,

$$\sum_{a_{1:H}} \int_{r_{1:H} \in \mathcal{E}_b} |r_{t_0}| \left| \Pi_{t=1}^H p_m(a_t, r_t) - \Pi_{t=1}^H g_m(a_t, r_t) \right| \pi(a_{1:H}|r_{1:H-1})d(r_{1:H})$$

$$= \sum_{a_{1:H}} \int_{r_{1:H} \in \mathcal{E}_b} |r_{t_0}| \cdot \Pi_{t=1}^H g_m(a_t, r_t)\pi(a_{1:H}|r_{1:H-1})d(r_{1:H})$$

$$\leq \sum_{a_{1:H}} \int_{r_{1:H} \in \mathcal{E}_b \cap \{|r_{t_0}| \leq z_Z\}} |r_{t_0}| \cdot \Pi_{t=1}^H g_m(a_t, r_t)\pi(a_{1:H}|r_{1:H-1})d(r_{1:H})$$

$$+ \sum_{a_{1:H}} \int_{r_{1:H} \in \mathcal{E}_b \cap \{|r_{t_0}| > z_Z\}} |r_{t_0}| \cdot \Pi_{t=1}^H g_m(a_t, r_t)\pi(a_{1:H}|r_{1:H-1})d(r_{1:H})$$

$$\leq z_Z \cdot \mathbb{P}_m(\mathcal{E}_b) + \sum_{a_{1:t_0}} \int_{r_{1:t_0} \in \{|r_{t_0}| > z_Z\}} |r_{t_0}| \cdot \Pi_{t=1}^{t_0} g_m(a_t, r_t)\pi(a_{1:t_0}|r_{1:t_0-1})d(r_{1:t_0}),$$

where the last inequality results from integrating over probabilities for time steps $t_0 + 1, ..., H$. The last summation term can be further bounded by integrating out $t_0^{\text{th}}$ time step since

$$\int_{\{|r_{t_0}| > z_Z\}} |r_{t_0}| g_m(a_{t_0}, r_{t_0})d(r_{t_0}) = \int_{\{|x| > 4\sqrt{\log(H/\epsilon)}\}} \frac{|x|}{\sqrt{2\pi}} \exp\left(-\frac{(x - \mu_m(a_t))^2}{2}\right) dx$$

$$\leq 2\mathbb{E}_{X \sim \mathcal{N}(0,1)}\left[|X| \cdot \mathbb{1}\{|X| > 3\sqrt{\log(H/\epsilon)}\}\right] \leq 2\epsilon^2/H^2,$$

where in the first inequality we used $|\mu_m(a_t)| \leq 1 \leq \sqrt{\log(H/\epsilon)}$. In last inequality we used $\mathbb{E}[|X| \cdot \mathbb{1}\{|X| \geq t\}] \leq \sqrt{\mathbb{P}(|X| \geq t)}$ by Cauchy-Schwartz inequality, and then used the Gaussian tail bound $\mathbb{P}(|X| \geq t) \leq \exp(-t^2/2)$ for $t = 3\sqrt{\log(H/\epsilon)}$. Therefore, we now get

$$\sum_{a_{1:H}} \int_{r_{1:H} \in \mathcal{E}_b} |r_{t_0}| \left| \Pi_{t=1}^H p_m(a_t, r_t) - \Pi_{t=1}^H g_m(a_t, r_t) \right| \pi(a_{1:H}|r_{1:H-1})d(r_{1:H})$$

$$\leq 4\sqrt{\log(H/\epsilon)} \cdot \epsilon^4/H^3 + 2\epsilon^2/H \leq 4\epsilon^2/H,$$

where we used $\mathbb{P}_m(\mathcal{E}_b) \leq H \cdot \mathbb{P}(|X| \geq z_Z - 1) \leq \epsilon^4/H^3$ with sufficiently small $\epsilon > 0$. This can be similarly done for all $t_0$, and thus

$$\sum_{a_{1:H}} \int_{r_{1:H} \in \mathcal{E}_b} \left(\sum_{t=1}^H |r_t|\right) \left| \Pi_{t=1}^H p_m(a_t, r_t) - \Pi_{t=1}^H g_m(a_t, r_t) \right| \pi(a_{1:H}|r_{1:H-1})d(r_{1:H}) \leq 4\epsilon^2.$$

We remain to bound

$$\sum_{a_{1:H}} \int_{r_{1:H} \in \mathcal{E}_b^c} \left(\sum_{t=1}^H |r_t|\right) \left| \Pi_{t=1}^H p_m(a_t, r_t) - \Pi_{t=1}^H g_m(a_t, r_t) \right| \pi(a_{1:H}|r_{1:H-1})d(r_{1:H}).$$

Note that $|r_t| < z_Z$ for all $t \in [H]$ when $r_{1:H} \in \mathcal{E}_b^c$. For each $t \in [H]$, we aim to bound

$$\sum_{a_{1:H}} \int_{r_{1:H} \in \mathcal{E}_b^c} |r_t| \left| \Pi_{t=1}^H p_m(a_t, r_t) - \Pi_{t=1}^H g_m(a_t, r_t) \right| \pi(a_{1:H}|r_{1:H-1})d(r_{1:H}).$$

Next, for any $r_t \in [z_s, z_{s+1}]$ for $s \in [L-1]$, we observe that

$$|p_m(a_t, r_t) - g_m(a_t, r_t)| = \frac{1}{z_{s+1} - z_s} \int_{z_s}^{z_{s+1}} |g_m(a_t, x) - g_m(a_t, r_t)| \, dx$$

$$\leq \frac{1}{z_{s+1} - z_s} \int_{z_s}^{z_{s+1}} |g_m(a_t, x)| + \frac{1}{z_{s+1} - z_s} \int_{z_s}^{z_{s+1}} |g_m(a_t, x) - g_m(a_t, r_t)| \, dx.$$

Then we observe that

$$|g_m(a_t, x) - g_m(a_t, r_t)| = \left| \frac{d}{dx} g_m(a_t, x')(r_t - x') \right|$$

$$\leq \frac{1}{\sqrt{2\pi}} |x' \exp(-x'^2/2)||z_{s+1} - z_s|,$$

for some $x' \in [z_s, z_{s+1}]$ where we used the mean-value theorem. A simple algebra shows that for any $x, x' \in [z_s, z_{s+1}]$,

$$|x' \exp(-x'^2/2) - x \exp(-x^2/2)| \leq |r_t \exp(-r_t^2/2)| + 2|z_{s+1} - z_s|,$$

where we used the second derivative of $x \exp(-x^2/2)$, which is $(x^2 - 1) \exp(-x^2/2)$, is always less than 1 in absolute value.

Plugging above relations into bounding the difference between $p_m$ and $g_m$ yields

$$|p_m(a_t, x) - g_m(a_t, r_t)| \leq \frac{1}{z_{s+1} - z_s} \left( \frac{\epsilon^8}{H^8} \int_{z_s}^{z_{s+1}} g_m(a_t, x) dx + \int_{z_s}^{z_{s+1}} |g_m(a_t, x) - g_m(a_t, r_t)| dx \right)$$

$$\leq \frac{\epsilon^8}{H^8} + \frac{1}{\sqrt{2\pi}} \int_{z_s}^{z_{s+1}} |r_t \exp(-r_t^2/2)| + 2|z_{s+1} - z_s| dx$$

$$\leq \frac{3\epsilon^2}{H^4} + \frac{2\epsilon}{H^2 \sqrt{2\pi}} |r_t| \cdot \exp(-r_t^2/2).$$

Using this the above, we bound

$$\sum_{a_{1:H}} \int_{r_{1:H} \in \mathcal{E}_b^c} |r_t| \left| \Pi_{t=1}^H p_m(a_t, r_t) - \Pi_{t=1}^H g_m(a_t, r_t) \right| \pi(a_{1:H}|r_{1:H-1}) d(r_{1:H}). \tag{24}$$

If $t = H$, then

$$(24) \leq \sum_{a_{1:H}} \int_{r_{1:H-1}} \Pi_{t=1}^{H-1} g_m(a_t, r_t) \pi(a_{1:H}|r_{1:H-1}) d(r_{1:H-1}) \int_{r_H \in \mathcal{E}_b^c} |r_H||p_m(a_H, r_H) - g_m(a_H, r_H)| \cdot d(r_H)$$

$$+ \sum_{a_{1:H}} \int_{r_{1:H-1} \in \mathcal{E}_b^c} \left| \Pi_{t=1}^H p_m(a_t, r_t) - \Pi_{t=1}^H g_m(a_t, r_t) \right| \pi(a_{1:H}|r_{1:H-1}) d(r_{1:H-1}) \int_{r_H \in \mathcal{E}_b^c} |r_H| p_m(a_H, r_H) \cdot d(r_H).$$

For the first term,

$$\int_{r_H \in \mathcal{E}_b^c} |r_H||p_m(a_H, r_H) - g_m(a_H, r_H)| \cdot d(r_H) \leq z_Z \cdot (z_Z - z_1) \cdot \frac{3\epsilon^2}{H^4} + \frac{4\epsilon}{H^2} \leq \frac{8\epsilon}{H^2},$$

where we used $\int_{r_H} \frac{1}{\sqrt{2\pi}} r_H^2 \exp(-r_H^2/2) \leq 1$ and $32(\epsilon/H) \cdot \log(H/\epsilon) < 1$ for sufficiently small $\epsilon$.
Note that we also have

$$\sum_{a_{1:H}} \int_{r_{1:H-1}} \Pi_{t=1}^{H-1} g_m(a_t, r_t) \pi(a_{1:H}|r_{1:H-1}) d(r_{1:H-1}) \leq 1.$$

For the second term, we first have

$$\int_{r_H \in \mathcal{E}_b^c} |r_H| p_m(a_H, r_H) \cdot d(r_H) \leq 1 + \mu_m(a_H) \leq 2.$$

Furthermore, we can show that

$$\sum_{a_{1:H}} \int_{r_{1:H-1} \in \mathcal{E}_b^c} \left| \Pi_{t=1}^H p_m(a_t, r_t) - \Pi_{t=1}^H g_m(a_t, r_t) \right| \pi(a_{1:H}|r_{1:H-1}) d(r_{1:H-1})$$

$$= \sum_{a_{1:H-1}} \int_{r_{1:H-1} \in \mathcal{E}_b^c} \left| \Pi_{t=1}^H p_m(a_t, r_t) - \Pi_{t=1}^H g_m(a_t, r_t) \right| \pi(a_{1:H-1}|r_{1:H-2}) \sum_{a_H} \pi(a_H|a_{1:H-1}, r_{1:H-1}) d(r_{1:H-1})$$

$$= \sum_{a_{1:H-1}} \int_{r_{1:H-1} \in \mathcal{E}_b^c} \left| \Pi_{t=1}^H p_m(a_t, r_t) - \Pi_{t=1}^H g_m(a_t, r_t) \right| \pi(a_{1:H-1}|r_{1:H-2}) d(r_{1:H-1}),$$

from which we recursively apply similar arguments. Thus we can conclude that

$$\sum_{a_{1:H}} \int_{r_{1:H} \in \mathcal{E}_b^c} |r_H| \left| \Pi_{t=1}^H p_m(a_t, r_t) - \Pi_{t=1}^H g_m(a_t, r_t) \right| \pi(a_{1:H}|r_{1:H-1}) d(r_{1:H}) \leq O(\epsilon/H).$$

$|r_t|$ with other time steps can also be similarly bounded. Thus, we can conclude that

$$\sum_{a_{1:H}} \int_{r_{1:H}} \left( \sum_{t=1}^H |r_t| \right) \left| \Pi_{t=1}^H p_m(a_t, r_t) - \Pi_{t=1}^H g_m(a_t, r_t) \right| \pi(a_{1:H}|r_{1:H-1}) d(r_{1:H}) \leq O(\epsilon),$$

and therefore $|V(\pi) - \tilde{V}(\pi)| \leq O(\epsilon)$ since $\sum_{m=1}^M w_m = 1$.

## Appendix E  Deferred Details in Section 4

### E.1  Additional Definitions

Let us define a few notation and interaction protocol. Suppose at the beginning of episode, a latent context $m_0 \in [M]$ is chosen, and and at each time step $t \in [H]$, we play $a_{i_t}$ where $i_t$ is sampled from $\text{Unif}([n])$. Let $\boldsymbol{i} = (i_1, i_2, ..., i_H)$ be the sequence of indices of played core actions, and $\boldsymbol{b} = (b_1, b_2, ...b_H)$ be the event-observation sequence in the episode, where $b_t := \mathbb{1}\{r_t = Z_{i_t}\}$. Let the parameter space $\Theta$ be the set of valid parameters:

$$\Theta = \{\theta = \{(w_m, \nu_m)\}_{m=1}^M | \forall j \in [n], m \in [M] \text{ s.t. } w_m, \nu_m(i) \in \mathbb{R}_+, \sum_{m=1}^M w_m = 1, \nu_m(j) \leq 1\}. \tag{25}$$

We use superscript $k$ to denote quantities observed in the $k^{\text{th}}$ episode. The probability of a trajectory under a model $\theta \in \Theta$ in the $k^{\text{th}}$ episode is defined by

$$\mathbb{P}_\theta(\boldsymbol{b}^k, \boldsymbol{i}^k) := (1/n)^H \cdot \sum_{m=1}^M w_m \Pi_{t=1}^H (b_t^k \nu_m(i_t) + (1 - b_t^k)(1 - \nu_m(i_t))).$$

### E.2  Polynomial Upper Bounds with Separation

In this subsection, we specify the details on separation conditions that make the polynomial sample complexity possible with MLE solutions. Suppose that there exists a context revealing action for any $m \neq m' \in [M]$, *i.e.*, we are given the following assumption:

**Assumption E.1 (Separated Bandit Instances)** *For any $m \neq m' \in [M]$, there exists some (unknown) $a \in \mathcal{A}$ such that $\|\mu_m^*(a, \cdot) - \mu_{m'}^*(a, \cdot)\|_1 \geq \gamma$ for some known $\gamma > 0$.*

Under Assumption E.1, if the time horizon $H = \tilde{O}(Z^2 M^2 / \gamma^2)$ is given enough to identify the context within each episode, then we can significantly improve the sample complexity for learning LMAB, from exponential to polynomial. Note that the time-horizon $H$ can be still much smaller than $A$ and thus we cannot explore all actions within a single episode, which is in contrast to explicit clustering based approaches studied in [9, 16].

Maximum likelihood estimator for LMABs with separation can guarantee the following:

**Lemma E.2** *Consider the maximum likelihood estimator $\theta_N = \{(\hat{w}_m, \hat{\nu}_m)\}_{m=1}^M$ under Assumption E.1 with time-horizon $H \geq C_1 \cdot nMZ^2 \log(1/(\epsilon w_{\min}))/\gamma^2$ for some universal constant $C_1 > 0$. If $N = C_2 \cdot w_{\min}^{-2} n \cdot \log(N/\eta)/\epsilon^2$ for some large constant $C_2 > 0$, then with probability at least $1 - \eta$, we have (up to some permutations in $\theta_N$)*

$$|w_m^* - \hat{w}_m| \leq \epsilon w_{\min}, \ \|\nu_m^* - \hat{\nu}_m\|_\infty \leq 2\epsilon, \qquad \forall m \in [M].$$

To get the above result, we first observe a consequence due to experimental design: the converse of Corollary 3.2 implies that if Assumption E.1 holds for all $m \neq m'$, then we have $\|\nu_m^* - \nu_{m'}^*\|_\infty \geq$

$\gamma/(Z\sqrt{2M})$. Thus, if $H/n = \tilde{O}(MZ^2/\gamma^2)$, then we can play each core action $O(H/n)$-times and get an $O(\gamma/(Z\sqrt{M}))$-accurate estimator $\hat{\nu}_m$ for one of $\{\nu_{m'}^*\}_{m'=1}^M$. Since we have good separation between samples from different contexts, by proper clustering arguments, the sample complexity of recovering $\theta^*$ can be polynomial. With Lemma E.2, we can use equation (4) to connect the near-optimality of returned policy computed with $\{(\hat{w}_m, \hat{\mu}_m)\}_{m=1}^M$ and the closeness in Wasserstein metric. We mention that for $H$, the dependence on $Z$ can be removed with more computationally expensive experimental design (see also Remark C.1).

### E.3 Proof of Lemma 4.1

We connect the maximum likelihood estimator to total variation distance between observations from $\theta^*$ and $\theta_N$. The connection between MLE $\theta_N$ and closeness in distributions of observations $b$ can be established by the following lemma.

**Lemma E.3** *There exists a universal constant $C > 0$ such that with probability at least $1 - \eta$,*

$$\sum_{b \in \{0,1\}^H} \sum_{i \in [n]^H} |\mathbb{P}_{\theta_N}(b, i) - \mathbb{P}_{\theta^*}(b, i)| \le C\sqrt{\frac{n\log(nHN) + \log(1/\eta)}{N}}.$$

That is, total variation distance between two observation distributions is bounded by $\tilde{O}\left(\sqrt{n/N}\right)$. On the other hand, for any $l \in [\min(H, 2M-1)]$ and any multi-index $(i_1, i_2, ..., i_l) \in [n]^l$, we have

$$C\sqrt{\frac{n\log(nHN) + \log(1/\eta)}{N}} \ge \sum_{b,i} |\mathbb{P}_{\theta_N}(b, i) - \mathbb{P}_{\theta^*}(b, i)|$$

$$\ge \sum_b |\mathbb{P}_{\theta_N} - \mathbb{P}_{\theta^*}|(b_{1:l}|a_{1:l} = (a_{i_1}, a_{i_2}, ..., a_{i_l})) \cdot \mathbb{P}(a_{1:d} = (a_{i_1}, a_{i_2}, ..., a_{i_l}))$$

$$= \sum_{b_{1:l}} |\mathbb{P}_{\theta_N} - \mathbb{P}_{\theta^*}|(b_{1:l}|a_{1:l} = (a_{i_1}, a_{i_2}, ..., a_{i_l})) \cdot n^{-l} \ge n^{-l} \cdot \|\hat{T}_l - T_l\|_\infty.$$

This implies that

$$\|\hat{T}_l - T_l\|_\infty \le Cn^l \cdot \sqrt{\frac{n\log(HNn) + \log(1/\eta)}{N}}.$$

Applying this to all $l \in [\min(H, 2M-1)]$, we get Lemma 4.1.

### E.4 Proof of Lemma E.2

From Lemma E.3, without loss of generality, we assume that the total variation distance between observations from $\theta_N$ and $\theta^*$ is bounded by $\epsilon w_{\min}/2$ since $N = \tilde{O}(w_{\min}^{-2} n/\epsilon^2)$:

$$\sum_{b,i} |\mathbb{P}_{\theta_N}(b, i) - \mathbb{P}_{\theta^*}(b, i)| \le \epsilon w_{\min}/2.$$

We will verify that for every $m \in [M]$, there exists $m' \in [M]$ such that $\|\hat{\nu}_m - \nu_{m'}^*\|_\infty \le 2\epsilon$ and $|\hat{w}_m - w_{m'}^*| \le \epsilon w_{\min}$.

First note that for all $m \ne m' \in [M]$, we have $\|\nu_m^* - \nu_{m'}^*\|_\infty \ge \lambda := \gamma/(Z\sqrt{2M})$. If it is not, then the model does not satisfy Assumption E.1. We start with the following lemma:

**Lemma E.4** *Suppose that there exists $m \in [M]$ such that $\|\nu_m^* - \hat{\nu}_{m'}\|_\infty \ge \lambda/4$ for all $m' \in [M]$. For every $j \in [n]$, define $E_{m,j}$ an event defined as:*

$$E_{m,j} := \left\{ \left| \frac{\sum_{t=1}^H \mathbb{1}\{i_t = j\} b_t}{\sum_{t=1}^H \mathbb{1}\{i_t = j\}} - \nu_m^*(j) \right| < \lambda/8 \right\},$$

*and let $E_m = \cap_{j=1}^n E_{m,j}$. Then*

$$\mathbb{P}_{\theta^*}(E_m) \ge w_m/2, \ \mathbb{P}_{\theta_N}(E_m) \le \epsilon w_{\min}. \tag{26}$$

*Proof.* Let us first check that $\mathbb{P}_{\theta^*}(E) \geq w_m/2$. Let $n_j = \sum_{t=1}^H \mathbb{1}\{i_t = j\}$. Since,

$$\mathbb{P}_{\theta^*}(E) \geq w_m \cdot \mathbb{P}_{\theta^*}(E|m_0 = m) = w_m \cdot \cap_{j=1}^n \mathbb{P}_{\theta^*}(E_j|m_0 = m),$$

it suffices to show that

$$\mathbb{P}_{\theta^*}(E_j|m_0 = m) \geq \mathbb{P}_{\theta^*}\left(\left|\sum_{t=1}^H \mathbb{1}\{i_t = j\} b_t - n_j \nu_m^*(j)\right| \leq n_j \lambda/8 \;\middle|\; m_0 = m, n_j \geq H/(2n)\right) \mathbb{P}_{\theta^*}(n_j \geq H/(2n))$$

$$\geq \left(1 - \exp(-\lambda^2 \cdot H/(64n))\right)\left(1 - \exp\left(\frac{-(1/2)(H/2n)^2}{H(1/n)(1-1/n) + (1/3)(H/2n)}\right)\right).$$

where in the last inequality we applied Hoeffeding's concentration inequality for first term, and then used Bernstein's inequality. Plugging $H > C \cdot nMZ^2 \log(1/(\epsilon w_{\min}))/\gamma^2$ for some sufficiently large constant $C > 0$, we get

$$\mathbb{P}_{\theta^*}(E_j|m_0 = m) \geq \left(1 - \exp(-2(\gamma^2/(128Z^2M)) \cdot H/(2n))\right) \cdot (1 - \exp(-H/(16n))) \geq 1 - (\epsilon w_{\min})^2.$$

By union bound, we have $\mathbb{P}_{\theta^*}(E) \geq w_m \cdot (1 - n\epsilon^2 w_{\min}^2) \geq w_m/2$.

Now we check that $\mathbb{P}_{\theta_N}(E) \leq \epsilon w_{\min}$. Starting from

$$\mathbb{P}_{\theta_N}(E) = \sum_{m'=1}^M w_{m'} \mathbb{P}_{\theta_N}(E|m_0 = m'),$$

it suffices to show that $\mathbb{P}_{\theta_N}(E|m_0 = m') \leq \epsilon w_{\min}$ for all $m' \in [M]$. Let us fix $m'$ and define

$$E_{m',j} := \left\{\left|\frac{\sum_{t=1}^H \mathbb{1}\{i_t = j\} b_t}{\sum_{t=1}^H \mathbb{1}\{i_t = j\}} - \hat{\nu}_m(j)\right| < \lambda/8\right\},$$

and let $E_{m'} = \cap_{j=1}^n E_{m',j}$. Following the same argument for $\mathbb{P}_{\theta^*}(E_j|m_0 = m)$, we can show that

$$\mathbb{P}_{\theta_N}(E_{m'}|m_0 = m') \geq 1 - (\epsilon w_{\min})^2.$$

If this happens, then it implies $E^c$ since $\|\nu_m^* - \hat{\nu}_{m'}\|_\infty \geq \lambda/2$. Thus,

$$\sum_{m'=1}^M \hat{w}_{m'} \mathbb{P}_{\theta_N}(E_{m'}|m_0 = m') \leq \sum_{m'=1}^M \hat{w}_{m'} \mathbb{P}_{\theta_N}(E^c|m_0 = m') = \mathbb{P}_{\theta_N}(E^c).$$

In other words, we have

$$\mathbb{P}_{\theta_N}(E) \leq 1 - \sum_{m'=1}^M \hat{w}_{m'} \mathbb{P}_{\theta_N}(E_{m'}|m_0 = m') \leq \epsilon^2 w_{\min}^2.$$

$\square$

The conclusion of Lemma E.4 contradict that $|\mathbb{P}_{\theta^*}(E_m) - \mathbb{P}_{\theta_N}(E_m)| \leq \epsilon w_{\min}/2$ due to the total variation distance bound from Lemma E.3. Thus, we ensure that for all $m \in [M]$, there exists $m' \in [M]$ such that $\|\nu_m^* - \hat{\nu}_{m'}\|_\infty \leq \lambda/4$.

Now without loss of generality, we can ignore the permutation invariance of models and assume that

$$\|\nu_m^* - \hat{\nu}_m\|_\infty \leq \lambda/4, \qquad \forall m \in [M].$$

Then we now show that for all $m \in [M]$, it holds that

$$|w_m^* - \hat{w}_m^*| \leq \epsilon w_{\min}, \quad \|\nu_m^* - \hat{\nu}_m^*\|_\infty \leq \epsilon.$$

For every $m$, let us define $E_m$ similarly to Lemma E.4:

$$E_{m,j} := \left\{\left|\frac{\sum_{t=1}^H \mathbb{1}\{i_t = j\} b_t}{\sum_{t=1}^H \mathbb{1}\{i_t = j\}} - \nu_m^*(j)\right| < \lambda/2\right\},$$

and $E_m := \cap_{j=1}^n E_{m,j}$. Then we proceed as the following:

$$\sum_{\boldsymbol{b},\boldsymbol{i}} |\mathbb{P}_{\theta_N}(\boldsymbol{b},\boldsymbol{i}) - \mathbb{P}_{\theta^*}(\boldsymbol{b},\boldsymbol{i})| \geq \sum_{(\boldsymbol{i},\boldsymbol{b}) \in E_m} |\mathbb{P}_{\theta_N}(\boldsymbol{b},\boldsymbol{i}) - \mathbb{P}_{\theta^*}(\boldsymbol{b},\boldsymbol{i})|$$

$$= \sum_{(\boldsymbol{i},\boldsymbol{b}) \in E_m} \left| \sum_{m'=1}^{M} \hat{w}_{m'} \mathbb{P}_{\theta_N}(\boldsymbol{b},\boldsymbol{i}|m_0 = m') - \sum_{m'=1}^{M} w_{m'} \mathbb{P}_{\theta^*}(\boldsymbol{b},\boldsymbol{i}|m_0 = m') \right|$$

$$\geq \sum_{\boldsymbol{i}} \sum_{\boldsymbol{b}} |\hat{w}_m \mathbb{P}_{\theta_N}(\boldsymbol{b},\boldsymbol{i}|m_0 = m) - w_m \mathbb{P}_{\theta^*}(\boldsymbol{b},\boldsymbol{i}|m_0 = m)|$$

$$- (\hat{w}_m \mathbb{P}_{\theta_N}(E_m^c|m_0 = m) + w_m \mathbb{P}_{\theta^*}(E_m^c|m_0 = m))$$

$$- \sum_{m' \neq m} (\hat{w}_{m'} \mathbb{P}_{\theta_N}(E_m|m_0 = m') + w_{m'} \mathbb{P}_{\theta^*}(E_m|m_0 = m')).$$

Recall that for any $m' \neq m$, the event $E_m$ implies $E_{m'}^c$ since every $\nu_m^*$ and $\nu_{m'}^*$ are separated by at least $\lambda = \gamma/(Z\sqrt{2M})$, which implies that $\nu_m^*$ and $\hat{\nu}_{m'}$ are separated by $(3/4)\lambda$. Following the same argument as in the proof for Lemma E.4,

$$\mathbb{P}_{\theta_N}(E_m|m_0 = m') \leq \mathbb{P}_{\theta_N}(E_{m'}^c|m_0 = m') \leq \epsilon^2 w_{\min}^2,$$

$$\mathbb{P}_{\theta^*}(E_m|m_0 = m') \leq \mathbb{P}_{\theta^*}(E_{m'}^c|m_0 = m') \leq \epsilon^2 w_{\min}^2.$$

Similarly, we can also check that

$$\mathbb{P}_{\theta_N}(E_m^c|m_0 = m) \leq \epsilon^2 w_{\min}^2,$$

$$\mathbb{P}_{\theta^*}(E_m^c|m_0 = m) \leq \epsilon^2 w_{\min}^2.$$

Thus, we have

$$\sum_{\boldsymbol{i},\boldsymbol{b}} |\mathbb{P}_{\theta_N}(\boldsymbol{b},\boldsymbol{i}) - \mathbb{P}_{\theta^*}(\boldsymbol{b},\boldsymbol{i})| \geq \sum_{\boldsymbol{b},\boldsymbol{i}} |\hat{w}_m \mathbb{P}_{\theta_N}(\boldsymbol{b},\boldsymbol{i}|m_0 = m) - w_m \mathbb{P}_{\theta^*}(\boldsymbol{b},\boldsymbol{i}|m_0 = m)| - 2\epsilon^2 w_{\min}^2.$$

We remain to lower bound $\sum_{\boldsymbol{b},\boldsymbol{i}} |\hat{w}_m \mathbb{P}_{\theta_N}(\boldsymbol{b},\boldsymbol{i}|m_0 = m) - w_m^* \mathbb{P}_{\theta^*}(\boldsymbol{b},\boldsymbol{i}|m_0 = m)|$. First note that

$$\epsilon w_{\min}/2 \geq \sum_{\boldsymbol{b},\boldsymbol{i}} |\hat{w}_m \mathbb{P}_{\theta_N}(\boldsymbol{b},\boldsymbol{i}|m_0 = m) - w_m^* \mathbb{P}_{\theta^*}(\boldsymbol{b},\boldsymbol{i}|m_0 = m)| \geq |\hat{w}_m - w_m^*|.$$

Thus we have $|\hat{w}_m - w_m^*| \leq \epsilon w_{\min}$. Now given this, we can proceed as

$$\epsilon w_{\min}/2 \geq \sum_{\boldsymbol{b},\boldsymbol{i}} |\hat{w}_m \mathbb{P}_{\theta_N}(\boldsymbol{b},\boldsymbol{i}|m_0 = m) - w_m^* \mathbb{P}_{\theta^*}(\boldsymbol{b},\boldsymbol{i}|m_0 = m)|$$

$$\geq w_m^* \cdot \sum_{\boldsymbol{b},\boldsymbol{i}} |\mathbb{P}_{\theta_N}(\boldsymbol{b},\boldsymbol{i}|m_0 = m) - \mathbb{P}_{\theta^*}(\boldsymbol{b},\boldsymbol{i}|m_0 = m)| - |\hat{w}_m - w_m^*|.$$

Now let $i_m := arg\max_{j \in [n]} |\nu_m^*(j) - \hat{\nu}_m(j)|$. Define an event $b_t$ being 1 at the first time $a_{i_m}$ is played:

$$F := \{b_t = 1, t = arg\min_{t' \in [H]} i_{t'} = i_m\}.$$

Note that since $H \gg n\log(1/(\epsilon w_{\min}))$, $i_m$ is played at least once with probability at least $1 - (\epsilon w_{\min})^2$. Then we can lower bound the total variation distance as

$$\sum_{\boldsymbol{b},\boldsymbol{i}} |\mathbb{P}_{\theta_N}(\boldsymbol{b},\boldsymbol{i}|m_0 = m) - \mathbb{P}_{\theta^*}(\boldsymbol{b},\boldsymbol{i}|m_0 = m)| \geq |\mathbb{P}_{\theta_N}(F|m_0 = m) - \mathbb{P}_{\theta^*}(F|m_0 = m)|$$

$$\geq |\nu_m^*(i_m) - \hat{\nu}_m(i_m)| - 2(\epsilon w_{\min})^2.$$

Hence we can conclude that

$$\epsilon w_{\min}/2 \geq w_m |\nu_m^*(i_m) - \hat{\nu}_m(i_m)| - 2(\epsilon w_{\min})^2 - \epsilon w_{\min},$$

which implies $\|\nu_m^* - \hat{\nu}_m\|_\infty \leq 2\epsilon$.

Note that the Wasserstein distance can thus be bounded as

$$W(\gamma^*, \hat{\gamma}) \leq \sum_{m=1}^{M} |w_m^* - \hat{w}_m| + \sum_{m=1}^{M} w_m^* \|\nu_m^* - \hat{\nu}_m\|_\infty \leq \epsilon M w_{\min} + 2\epsilon \leq 3\epsilon.$$

**Remark E.5** *In our guarantee, we assumed that $H \gg \log(1/\epsilon)$, i.e., $H$ should be increased logarithmically with the final accuracy. As one might imagine, this is not the optimal condition for separations between individual models. A more delicate and technically involved analysis might reveal that the sub-optimal dependency on $\log(1/\epsilon)$ can be dropped with the EM algorithm as in learning Gaussian mixture models (e.g., [22]). Since it is technically much more complicated, we leave the task of verifying more tight separation conditions as future work.*

## E.5 Proof of Lemma E.3

This is a rather standard consequence of MLE for parameterized distributions. Let us define well-conditioned parameters $\Theta' \subseteq \Theta$ defined as follows:

$$\Theta' = \Big\{ \theta = \{(w_m, \nu_m)\}_{m=1}^M | \forall j \in [n], m \in [M] \text{ s.t. } w_m, \ \nu_m(j) \in \mathbb{R}_+,$$

$$\sum_{m=1}^M w_m = 1,$$

$$\epsilon \leq w_m, \ \epsilon \leq \nu_m(j) \leq 1 - \epsilon, \quad \forall m \in [M], j \in [n] \Big\}.$$

Let $\Theta_\epsilon$ be $\epsilon^2$-covering of $\Theta'$. Since there are $n + M$ free parameters, log-cardinality of $\Theta_\epsilon$ is at most $O(1/\epsilon)^{2(n+M)}$. Note that $\Theta_\epsilon$ is a (not necessarily minimal) $\epsilon$-cover for $\Theta$ as well.

To simplify the notation, we often use $X := (\boldsymbol{b}, \boldsymbol{i})$ (and $X^k = (\boldsymbol{b}^k, \boldsymbol{i}^k)$) to replace a sample trajectory. Our goal is to bound

$$TV(\theta_N, \theta^*) := \sum_{X=(\boldsymbol{b}, \boldsymbol{i}): \boldsymbol{b} \in \{0,1\}^H, \boldsymbol{i} \in [n]^H} |\mathbb{P}_{\theta_N}(X) - \mathbb{P}_{\theta^*}(X)|,$$

where $TV(\theta_1, \theta_2)$ is a total variation distance between $\mathbb{P}_{\theta_1}$ and $\mathbb{P}_{\theta_2}$ for any $\theta_1, \theta_2 \in \Theta$.

Let $\overline{\theta} = \{(\overline{w}_m, \overline{\nu}_m\}_{m=1}^M \in \Theta_\epsilon$ such that

$$\frac{w_m^*}{\overline{w}_m} \leq 1 + 2\epsilon, \qquad\qquad \forall m \in [M],$$

$$\frac{\nu_m^*(i)}{\overline{\nu}_m(i)} \leq 1 + 2\epsilon, \qquad\qquad \forall m \in [M], i \in [n],$$

$$\frac{1 - \nu_m^*(i)}{1 - \overline{\nu}_m(i)} \leq 1 + 2\epsilon, \qquad\qquad \forall m \in [M], i \in [n].$$

Such $\overline{\theta}$ is guaranteed to exist in $\Theta_\epsilon$ by construction. A simple algebra shows that for any trajectory $X$, we have

$$\frac{\mathbb{P}_{\theta^*}(X)}{\mathbb{P}_{\overline{\theta}}(X)} \leq (1 + 2\epsilon)^H.$$

As long as $\epsilon < 1/H^2$, this is bounded by constant.

Now, fix any $\theta \in \Theta_\epsilon$ and let $l(X) := \frac{1}{2} \log\left(\frac{\mathbb{P}_\theta(X)}{\mathbb{P}_{\overline{\theta}}(X)}\right)$. Then using Chernoff's method, we get

$$\mathbb{P}_{\theta^*}\left(\sum_{k=1}^N l(X^k) - \log\left(\mathbb{E}_{\theta^*}\left[\exp\left(\sum_{k=1}^N l(X^k)\right)\right]\right) > \lambda\right)$$

$$= \mathbb{P}_{\theta^*}\left(\exp\left(\sum_{k=1}^N l(X^k) - \log\left(\mathbb{E}_{\theta^*}\left[\sum_{k=1}^N \exp(l(X^k))\right]\right)\right) > \exp(\lambda)\right)$$

$$\leq \mathbb{E}_{\theta^*}\left[\exp\left(\sum_{k=1}^N l(X^k) - \log\left(\mathbb{E}_{\theta^*}\left[\exp\left(\sum_{k=1}^N l(X^k)\right)\right]\right)\right)\right] \cdot \exp(-\lambda)$$

$$= \exp(-\lambda),$$

where we used $\mathbb{E}[\exp(S - \log(\mathbb{E}[\exp(S)]))] = \frac{\mathbb{E}[\exp(S)]}{\mathbb{E}[\exp(S)]} = 1$ for any $S$ and distribution. Taking union bound over $\Theta_\epsilon$, we can conclude that with probability at least $1 - \eta$,

$$-\log\left(\mathbb{E}_{\theta^*}\left[\Pi_{k=1}^N \sqrt{\frac{\mathbb{P}_\theta(X^k)}{\mathbb{P}_{\overline{\theta}}(X^k)}}\right]\right) \leq -\frac{1}{2}\sum_{k=1}^N \log\left(\frac{\mathbb{P}_\theta(X^k)}{\mathbb{P}_{\overline{\theta}}(X^k)}\right) + \log(|\Theta_\epsilon|/\eta).$$

Now, since $X^1, ..., X^N$ are independent, we have

$$-\log\left(\mathbb{E}_{\theta^*}\left[\Pi_{k=1}^N \sqrt{\frac{\mathbb{P}_\theta(X^k)}{\mathbb{P}_{\overline{\theta}}(X^k)}}\right]\right) = -\sum_{k=1}^N \log\left(\mathbb{E}_{\theta^*}\left[\sqrt{\frac{\mathbb{P}_\theta(X)}{\mathbb{P}_{\overline{\theta}}(X)}}\right]\right)$$

$$= -\sum_{k=1}^N \log\left(\mathbb{E}_{\theta^*}\left[\sqrt{\frac{\mathbb{P}_\theta(X)}{\mathbb{P}_{\theta^*}(X)}} \cdot \sqrt{\frac{\mathbb{P}_{\theta^*}(X)}{\mathbb{P}_{\overline{\theta}}(X)}}\right]\right)$$

$$\geq -\sum_{k=1}^N \log\left((1+2\epsilon)^{H/2} \cdot \mathbb{E}_{\theta^*}\left[\sqrt{\frac{\mathbb{P}_\theta(X)}{\mathbb{P}_{\theta^*}(X)}}\right]\right)$$

$$= -\frac{HN}{2}\log(1+2\epsilon) - \sum_{k=1}^N \log\left(\mathbb{E}_{\theta^*}\left[\sqrt{\frac{\mathbb{P}_\theta(X)}{\mathbb{P}_{\theta^*}(X)}}\right]\right).$$

Using $-\ln(x) \geq 1 - x$, we get

$$-\sum_{k=1}^N \log\left(\mathbb{E}_{\theta^*}\left[\sqrt{\frac{\mathbb{P}_\theta(X)}{\mathbb{P}_{\theta^*}(X)}}\right]\right) \geq \sum_{k=1}^N \left(1 - \mathbb{E}_{\theta^*}\left[\sqrt{\frac{\mathbb{P}_\theta(X)}{\mathbb{P}_{\theta^*}(X)}}\right]\right)$$

$$= \sum_{k=1}^N \left(1 - \sum_X \sqrt{\mathbb{P}_{\theta^*}(X)\mathbb{P}_\theta(X)}\right) = N\mathcal{H}^2(\theta^*, \theta),$$

where $\mathcal{H}(\theta_1, \theta_2)$ is a Hellinger distance between $\mathbb{P}_{\theta_1}$ and $\mathbb{P}_{\theta_2}$. Also note that $TV(\theta^*, \theta) \leq \mathcal{H}(\theta^*, \theta)$. Collecting all, we now can say that

$$N \cdot TV^2(\theta^*, \theta) \leq -\frac{1}{2}\sum_{k=1}^N \log\left(\frac{\mathbb{P}_\theta(X^k)}{\mathbb{P}_{\overline{\theta}}(X^k)}\right) + \log(\Theta_\epsilon/\eta) + \frac{HN}{2}\log(1+2\epsilon),$$

with probability at least $1 - \eta$. Finally, let $\overline{\theta}_N \in \Theta_\epsilon$ be the one close to $\theta_N$ such that

$$\frac{\mathbb{P}_{\theta_N}(X)}{\mathbb{P}_{\overline{\theta}_N}(X)} \leq (1+2\epsilon)^H,$$

similarly to when defining $\overline{\theta}$. Then,

$$N \cdot TV^2(\theta^*, \overline{\theta}_N) \leq -\frac{1}{2}\sum_{k=1}^N \log\left(\frac{\mathbb{P}_{\theta_N}(X^k)}{\mathbb{P}_{\overline{\theta}}(X^k)}\right) + \frac{1}{2}\sum_{k=1}^N \log\left(\frac{\mathbb{P}_{\theta_N}(X^k)}{\mathbb{P}_{\overline{\theta}_N}(X^k)}\right) + \log(\Theta_\epsilon/\eta) + \frac{HN}{2}\log(1+2\epsilon)$$

$$\leq \log(\Theta_\epsilon/\eta) + HN\log(1+2\epsilon),$$

where we used the fact that $\theta_N$ is the maximum likelihood estimator. With a proper scaling of $\epsilon \ll 1/(nHN)^4$, we get

$$TV(\theta^*, \overline{\theta}_N) \leq O\left(\sqrt{\frac{(n+M)\log(nHN) + \log(1/\eta)}{N}}\right).$$

Finally, it is not hard to show that $TV(\theta_N, \overline{\theta}_N) \leq 2\epsilon H \leq 1/N$. We can conclude that

$$TV(\theta^*, \theta_N) \leq C \cdot \sqrt{\frac{n\log(nHN) + \log(1/\eta)}{N}},$$

for some sufficiently large constant $C > 0$.