# OpenReview forum: "Tractable Optimality in Episodic Latent MABs"
_NeurIPS.cc/2022/Conference — NeurIPS 2022 Accept_

### Official Review · Reviewer_XPh1 · 2022-07-10

**Rating:** 7
**Confidence:** 3
**Soundness:** 3 good
**Presentation:** 4 excellent
**Contribution:** 4 excellent

**Summary:**

This paper studies the problem called Episodic Latent Multi-Armed Bandits, which includes $M$ MABs with $A$ actions. In each episode, an MAB is sampled with some probability, then the agent interacts with the MAB for $H$ steps, without knowing the true latent context. Previous works derived algorithms to learn a near-optimal policy with $O(A)^H$ episodes. This paper exploits the low-rank structure of the latent reward distributions and uses a moment-matching method to derive an algorithm with a $\mathrm{poly}(A)+\mathrm{poly}(H, M)^{\min \\{H, M \\}}$ sample complexity. The analysis depends on a finite, discrete support set of reward distributions, and can be extended to a Gaussian. This paper further provides a Maximum Likelihood implementation to approximate high-order moment matching and it shows relatively good results.

**Questions:**

None.

**Limitations:**

None.

**Strengths And Weaknesses:**

Strengths:
1. This paper gives nearly unconditional (without strong separability assumptions) results for LMAB problems, which are all polynomial sample complexities in $A$. This shows a separating gap between LMAB, Latent MDP, and reward-mixing MDPs.
2. To my knowledge, the use of the Kiefer-Wolfowitz theorem (Lemma 3.1 and Corollary 3.2) for dimension reduction with bandit feedback is novel.
3. This paper provides simulation to corroborate theories.
4. This paper is clearly written, with clear notations and logic.
5. The moment-matching technique may be further extended to LMDPs.


Weaknesses:
1. The assumption that either $H = O(1)$ or $M = O(1)$ is not negligible. I hope the authors can either provide a $\mathrm{poly}(H, M)^{\min \\{H, M \\}}$ lower-bound or further improve on this exponential term.
2. Some polynomial terms in Theorem 3.8 have large constant exponents. I don't know if this is tight or not. But according to the reward-mixing MDP paper [22], I guess they have room to be improved.

---

> ### Author Response · Authors · 2022-08-02
> **Comments to Reviewer XPh1**
>
> We thank the reviewer for their interest, for the feedback and the concise summary of our work!  We supply our comment to the remarks below.
>
> **Assumption of either $H=O(1)$ or $M=O(1)$ and minimax rate (polynomial terms and constant in the exponents).** We do not currently have a lower bound for the latent MAB setting. Kwon et al. 2021 established an exponential in $M$ lower bound for the broader class of latent MDPs, which may give some indication for the hardness of the more specialized latent MAB setting. That being said, establishing sample complexity lower bound and achieving the minimax performance is an important and interesting future research question. Indeed, we highlighted this in the summary section of our work.

---

> > ### Comment · Reviewer_XPh1 · 2022-08-05
> > **post rebuttal response**
> >
> > I thank the authors for their reply. The score is unchanged.

---

### Official Review · Reviewer_E6HY · 2022-07-11

**Rating:** 6
**Confidence:** 3
**Soundness:** 3 good
**Presentation:** 3 good
**Contribution:** 3 good

**Summary:**

This paper studies latent MABs with discrete reward. First, the paper suggested that in order to identify a near-optimal policy, it suffices to identify the high-order moments of the action-reward distribution, which essentially recover the distribution over all possible action-reward trajectory. Second, since this tensor is low rank, the paper proposed to decouple the estimation into three steps: first estimate the subspace, then use experimental design to find a core-set, and finally only need to estimate a sub high-order tensor corresponding to the core set. By doing so, they avoid the $A^{\mathcal{O}(\min\{M,H\})}$ factor in sample complexity.

**Questions:**

Please see my questions above.

**Strengths And Weaknesses:**

Strengths:

1. The result is almost assumption-free. Even if this paper only focuses on the finite MAB setting, it is nice to see they don't require any observable or revealing condition.

2. The algorithm design is cute. First estimate the subspace, and then instead of directly estimating the $\ell_2$ projection onto the subspace, they use the subspace to compute a $\ell_1$ core set of much smaller size than $AZ$.


Weakness:

1. It is hard for me to tell how significant it is to improve from $(AZ)^M$ to $Z^M$. If $M$ is large, the result obtained is still exponential in $M$. If $M$ is small, then $A^M$ is still fine?

2. I found the technical writing in Section 3 is a little bit over-detailed. For example, in Section 3.1, the paper spent much words motivating the core-set trick before giving an overview of the overall algorithm flow. Maybe give an algorithmic overview first and then explain each module in detail? Anyway this is just my own feeling and not an important issue.

---

> ### Author Response · Authors · 2022-08-02
> **Comments to Reviewer E6HY**
>
> Thanks for the comments and warm feedback! We will address the questions below.
>
> **Summary of our work.** Prior to discussing the comments, we wish to highlight that we also study the setting in which the reward distribution is continuous and Gaussian.  This shows that our techniques may be adapted to other parametric families with continuous reward distribution besides the finite and discrete reward setting.
>
>
> **(AZ)^M vs Z^M.** We agree with the reviewer that removing the $Z^M$ factor is highly desirable and we consider it to be a challenging question for future research. Nevertheless, when the size of the reward alphabet is small (i.e., $Z=O(1)$) and the number of actions is large there is a significant gap between ${(AZ)}^M$ vs $Z^M$ factors. We will further highlight this direction as a future research question in the final version. Thank you!
>
>
> **Writing suggestion.** Thanks for the suggestion. We will carefully review the way we presented our main idea and algorithm, and reflect the comment in the revision accordingly.

---

> > ### Comment · Reviewer_E6HY · 2022-08-08
> > **Re author response**
> >
> > Thanks for the response. I will leave my score unchanged.

---

### Official Review · Reviewer_rtQX · 2022-07-12

**Rating:** 5
**Confidence:** 2
**Soundness:** 3 good
**Presentation:** 3 good
**Contribution:** 3 good

**Summary:**

The paper studies episodic latent MABs, where the learner interacts with a sequence of bandit problems in episodes of H steps and each bandit problem is associated with a latent context m \in [M] which characterizes the arms' reward distributions. The authors present an algorithm based on moment matching that achieves a sample complexity polynomial in A and exponential in M or H, hence improving over a trivial reduction to a partially-observable MDP which would yield A^H complexity (at least in the regime where M or H is small).

**Questions:**

Overall I think the paper provides an interesting and significant contribution (though, as mentioned above, I am not very familiar with the related literature). I am thus more inclined to vote for acceptance. Here are a few additional comments/questions besides the strengths/weaknesses reported above.

1. While the episodic latent MAB setting introduces many challenges, I think it should be better motivated. For instance, in the recommender system example given in the first paragraph of the introduction, it is typically reasonable to assume that one knows some information about the user interacting with the system, in which case it is possible to use standard contextual-bandit strategies. Even when no information about the latent contexts is available, if H is large, one can simply run a no-regret bandit algorithm from scratch at each episode. While this would completely ignore the structure of the problem (and probably yield poor performance), it would result in a much simpler and tractable strategy. Could the authors characterize other relevant problems that fit into this formalism and where one really cares about competing with the optimal history-dependent policy (eg where running a contextual bandit algorithm is not possible)?

2. One of the main assumptions here is that the learner knows the number of contexts M. What if M is unknown? Would there be any hope to estimate it?

3. Is w_min, the minimum probability of sampling a context, assumed to be strictly positive? What if it is zero?

4. Another line of related works that should be probably mentioned is the one of "structured bandits" (see eg [1,2,3] below and references therein). As in the setting studied here, it is assumed that the arms' reward distributions dependent on some latent parameter. Differently from this paper, the learner knows the mappings but does not know the latent parameter

5. A minor comment on wording: a sample complexity of O(A^H) is still polynomial in A. So when stating the sample complexity as depending on poly(A), it would be good to clarify that the exponent of A is neither H nor M.

6. One downside of the approach is that it must explicitly distinguish between the large and small H regimes. Would it be possible to do a more unified treatment?


[1] Lattimore, Tor, and Rémi Munos. "Bounded regret for finite-armed structured bandits." Advances in Neural Information Processing Systems 27 (2014).
[2] Tirinzoni, Andrea, Alessandro Lazaric, and Marcello Restelli. "A novel confidence-based algorithm for structured bandits." International Conference on Artificial Intelligence and Statistics. PMLR, 2020.
[3] Degenne, Rémy, Han Shao, and Wouter Koolen. "Structure adaptive algorithms for stochastic bandits." International Conference on Machine Learning. PMLR, 2020.

**Limitations:**

The authors discussed the limitations of their work. I do not see any societal impact.

**Strengths And Weaknesses:**

Strengths:
- Novelty: the method proposed here, its analysis, and the results are, to my knowledge, novel (though I am not super familiar with the related literature on latent bandits/MDPs)
- The main result seems interesting and significant as it advances our understanding of the episodic latent MAB setting (in particular by confirming that the latter is provably easier than latent MDPs)
- While notation is quite complicated and there are a lot of technical details, I think the paper is very well written. In particular, the authors provide both intuitive and formal explanations of their results and clearly discuss the results while properly comparing them with related literature

Weaknesses:
- While the problem studied here is definitely very complicated and requires quite technical methods to be solved, I feel that the setting itself should be better motivated (see below)
- The results are only for rewards with finite support or Gaussian distributions and it is not clear whether they extend beyond that (eg to general sub-Gaussian rewards). This might limit the overall scope
- The "theoretical" algorithm based on tensor decomposition seems to be quite computationally inefficient. While the authors propose a simpler maximum-likelihood implementation, that still requires solving a non-convex optimization problem for which one has to rely on heuristics (hence loosing theoretical guarantees).
- While the main sample complexity result improves over the naive A^H baseline and advances our understanding of the problem at hand, it is still exponential in either M or H. Since no lower bound is provided, one still wonders whether such an exponential dependence is unavoidable
- While some numerical simulations have been reported in appendix, I feel that they should be given more importance and be moved to the main text

---

> ### Author Response · Authors · 2022-08-02
> **Comments to Reviewer rtQX**
>
> We appreciate the detailed and thoughtful comments made by the reviewer. We will address the main ones point-by-point.
>
> **On reward distribution.** We note that the optimal (history-dependent) policy requires the knowledge of the reward distribution, and thus we need to learn the reward distributions (this is one main difference from standard MAB where only the mean of rewards is required). Discrete and Gaussian distributions are canonical examples (of finite-support and continuous distributions, respectively), but we believe that our approach can be easily extended to a larger class of distributions. We will add discussions on this point in the final revision.
>
> **Computational issues.** We would like to clarify that we do not claim a computationally efficient algorithm for solving this problem, but instead we focus on the statistical aspect of the problem. Computational issues are very important and challenging on their own, and thus we plan to investigate them in future work. Despite the lack of computational guarantees, we demonstrated the effectiveness of the MLE approach empirically (section 4).
>
> **Implementing MAB algorithms.** Sequential decision problems with latent context give rise to interesting challenges. As previous work showed (Kwon, et al., 2021), there is substantial difference between the scenario in which the latent context is given to the learner at the end of each episode to the case where it is not given; for the first polynomial sample complexity is possible, whereas, for the latter, there is a lower bound which is exponential in $M$.
>
> When either $H$ is sufficiently large or the learner has access to some prior information the latent context might be accessible at the end of each episode. Then, standard reinforcement learning algorithms can be implemented (e.g., model based approach in Kwon, et al., 2021). Our work focuses on the scenario in which such assumption doesn't hold; when there is a significant uncertainty in the identity of the latent context. Such a scenario arises when the horizon is small, or when the latent MAB problem cannot be easily separated.
>
> **Number of latent models $M$ is unknown.** As the reviewer pointed out, instead of knowing the true $M$, we may only have a proper upper bound $\bar{M}$: in such cases, the same guarantees hold with $M$ being replaced by $\bar{M}$. The point of $M$ being unknown is whether we can find a moment-matching instance with our (coarse) guess of $M$, i.e., if we cannot find a moment-matching instance, then it implies that the true $M$ is greater than the current guess. We will add a discussion on this point in the revision.
>
> **Is $w_{\mathrm{min}}$ assumed to be positive?** This is a great point. It is desirable that $w_{\mathrm{min}}$ is positive and in the order of $O(1/M)$ if all contexts are well balanced. However, as the reviewer pointed out, we may consider the limit that $w_{\mathrm{min}}$ goes to zero. In that case, as can be inferred from the definition of $\delta_{\rm sub}$ in Eq. (5), effectively $w_{\mathrm{min}}$ is $\epsilon/(ZM)^H$ (we will fix a typo here) when $H < 2M-1$. This is mainly because in the parameter unidentifiable regime, we give guarantees from moment-closeness, and when giving guarantees through moment closeness, the subspace has to be very accurate to approximate higher-order moments well. Nevertheless, we are not aware whether this is tight for the $H < 2M-1$ regime, and we find this to be an interesting question for future research. We will make this point explicit in the final revision.
>
> **Structured bandits.** Thanks for the comment and the references. We will add a discussion on this line of work. Note that in the structured bandits setting the interaction takes place within a single episode; the learner interacts with an unknown but fixed structured MAB problem. In our setting, a new MAB problem is being drawn at the end of each episode. This makes the challenges that arise in both settings quite different.
>
> **Unified treatment for large $H$ and large $M$.** These two settings have a fundamental difference: if $H\gg M$ latent model identification is possible, whereas for $H\ll M$ it is not. This gives some intuition as for why we believe these two regimes should be handled separately in terms of analysis. That being said, the algorithm we use in both these cases is the same–up to modifying the number of higher-order moments that are needed to be estimated.
>
> **Assumption of either $H=O(1)$ or $M=O(1)$.** We don't currently have a lower bound for the latent MAB setting. Kwon et al. 2021 established an exponential in $M$ lower bound for the broader class of latent MDPs, which may give some indication for the hardness of the more specialized latent MAB setting. That being said, establishing sample complexity lower bound and achieving the minimax performance is an important and interesting future research question. Indeed, we highlighted this in the conclusion section of our work.

---

### Official Review · Reviewer_RY7y · 2022-07-12

**Rating:** 6
**Confidence:** 2
**Soundness:** 3 good
**Presentation:** 2 fair
**Contribution:** 3 good

**Summary:**

The present paper considers the challenging policy learning problem of latent multi-armed bandits. The setting as follows. There are M latent contexts, and corresponding to each latent context there is a distribution over multi-armed bandit problems. The objective of the learner is to identify a policy for sequentially choosing actions in the context of a random instance of such a multi-armed bandit problem. Performance is quantified in terms of the cumulative regret over the horizon H, when averaged over a distribution on the set of M contexts. In the setting, both the individual bandit problems for each context, and the distribution over contexts are unknown a priori. The key question is how many episodes the learner must explore before an epsilon-optimal policy is identified. In particular, is this number of the same order of the cardinality of the set of all action sequences (i.e. A^H, where A is the number of actions and H is the horizon).
The present paper demonstrates that it is possible to learn an epsilon-optimal policy in poly(A)+ploy(H,M)^min(H,M) episodes.
A key insight is that when the number of latent contexts M is significantly less than the number of actions A then one can reduce the scope for unfavorable dependencies upon the number of actions by projecting onto the M dimensional space of reward vectors, for each contexts. This motivates an approach where the goal is efficiently project sequences of actions and rewards onto this space. This requires an interesting adaptation of the Keifer-Wolfowitz theorem. The key difficulty then lies in identifying the space itself.
The analysis considers two separate regimes. The first is where the horizon is at least twice the number of latent contexts. If this is the case then the dimension reduction approach is used to infer the context. In the second regime, the horizon is to small to identify the context. If this is the case then a more direct approach is taken which directly matches the Hth order moments.
The main result is Theorem 3.8 which gives a bound on the required sample complexity which is polynomial in A, the number of actions.
The paper also considers a more traditional maximum likelihood based variant of the methodology based on the EM algorithm, which performs well in practice.


**Questions:**

Would anything be lost in just assuming that all of the arm-dependent reward distributions are Bernoulli? It seems that this would simplify the presentation.

**Limitations:**

The primary limitations are discussed above in the “strengths and weaknesses” section.

**Strengths And Weaknesses:**

The paper addresses a very challenging and interesting problem, which seems well-motivated by practical applications. In particular, the setting goes substantially beyond the closely related area of learning priors in Bayesian multi-armed bandits. Moreover, the algorithmic approach based on dimensionality reduction and Keifer-Wolfowitz theory seems novel in this setting.
I think a major limitation with the paper is the quality of the writing. I found the paper quite difficult to read in many places. This is partly due to my lack of expertise in this area. That being said, I think there were also issues with the clarity of presentation. As an example, the abstract refers to “A”. This denotes the number of actions, but we don’t actually learn this until line 32 of the text. There are also various strange uses of brackets. As another example, consider Assumption 2.2. In particular, its not at all clear what Z=O(1) is with respect to at this stage.
I would also argue that Figure 1 gives relatively little value over and above Section 1.2. It would be preferable to use the space to include a figure on empirical results for Section 4.

---

> ### Author Response · Authors · 2022-08-02
> **Comments to Reviewer RY7y**
>
> We thank the reviewer for the comments and the thoughtful description of this work.
>
> **Writing suggestions.**
> Thanks for these comments, we will implement these changes and revise the full paper as well!
>
> **Bernoulli reward.** We chose to keep track of Z–the size of the alphabet of the reward distribution–to highlight to the reader the dependence of our result on this quantity. For Bernoulli reward, when Z=2, such dependence would be less transparent to the reader in our opinion.  Furthermore, we will change the notation Z=O(1) and highlight that our results are interesting in the regime that Z is a small constant.
>
> **Figure 1.** Although Figure 1 does not add additional information on top of Section 1.2 we believe it has merits–it is simpler to understand the positioning of our work relative to existing literature. That being said, in the final version, we will move other parts of the paper to the appendix and bring the main experimental results to the main paper. Thanks!

---

### Meta-Review · Area_Chair_Kf8v · 2022-08-26

**Recommendation:** Accept
**Confidence:** Certain

**Metareview:**

All the reviewers found the problem solved to be challenging, at least mathematically interesting (and most reviewers found the problem itself to merit study), and the authors’ algorithmic approach to be elegant and novel. One reviewer has doubts about the practical applicability/relevance of the latent MAB (LMAB) model. All things considered, I am inclined to believe that situations with a short horizon do practically motivate the latent MAB model. Regarding the LMAB model itself, I find that it is worthy of studying, especially under the very weak assumptions that authors require. The results achievable here are a welcome contrast to the lower bounds in the LMDP model, in particular by avoiding a need for $\\Omega(A^M)$ episodes, and so I think theoretically this also is an interesting model to consider. This is a strong contribution and a welcome addition to the proceedings. Please be sure to consider the various presentational issues mentioned by the reviewers in preparing the final version of this work.

**Award:**

No

---

### Decision · Program_Chairs · 2022-09-14

Accept